# PartSAM: A Scalable Promptable Part Segmentation Model Trained on Native 3D Data

**Zhe Zhu**[1], **Le Wan**[2], **Rui Xu**[3], **Yiheng Zhang**[4], **Honghua Chen**[5], **Zhiyang Dou**[3], **Cheng Lin**[6], **Yuan Liu**[2†], **Mingqiang Wei**[1†]

[1]Nanjing University of Aeronautics and Astronautics,
[2]Hong Kong University of Science and Technology,
[3]The University of Hong Kong,
[4]National University of Singapore,
[5]Lingnan University,
[6]Macau University of Science and Technology

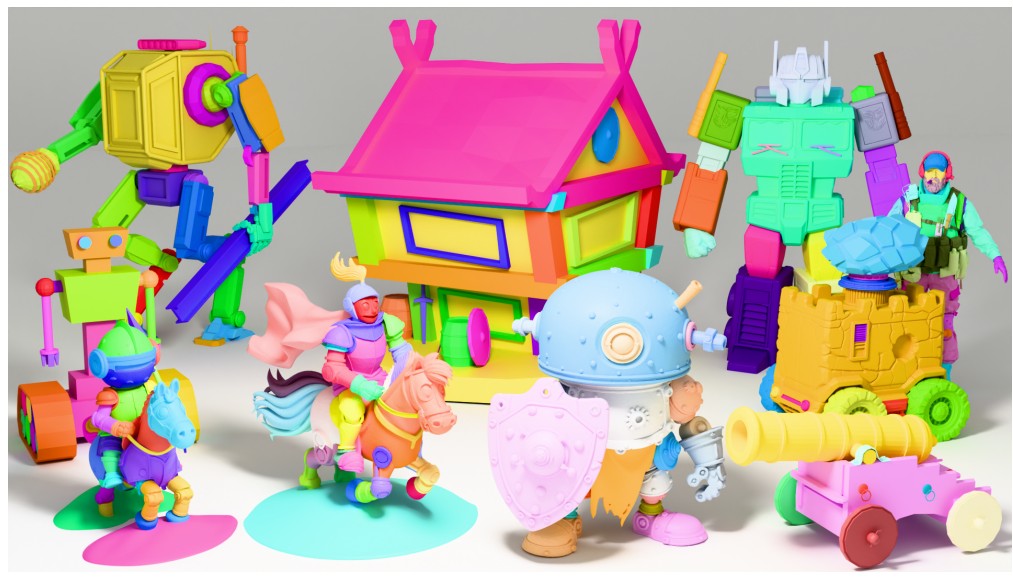

Figure 1: We propose PartSAM, a promptable 3D part segmentation model trained with large-scale native 3D data. The combination of a scalable architecture and large-scale training data endows PartSAM with strong generalization ability, enabling it to automatically decompose diverse 3D models, including both artist meshes and AI-generated shapes, into semantically meaningful parts.

## ABSTRACT

Segmenting 3D objects into parts is a long-standing challenge in computer vision. To overcome taxonomy constraints and generalize to unseen 3D objects, recent works turn to open-world part segmentation. These approaches typically transfer supervision from 2D foundation models, such as SAM, by lifting multi-view masks into 3D. However, this indirect paradigm fails to capture intrinsic geometry, leading to surface-only understanding, uncontrolled decomposition, and limited generalization. We present PartSAM, the first promptable part segmentation model trained natively on large-scale 3D data. Following the design philosophy of SAM, PartSAM employs an encoder–decoder architecture in which a triplane-based dual-branch encoder produces spatially structured tokens for scalable part-aware representation learning. To enable large-scale supervision, we further introduce a model-in-the-loop annotation pipeline that curates over five million 3D shape–part pairs from online assets, providing diverse and fine-grained labels. This combination of scalable architecture and diverse 3D data yields emergent open-world capabilities: with a single prompt, PartSAM achieves highly accurate part identifi-

---

[†]Corresponding authors.

cation, and in a "Segment-Every-Part" mode, it automatically decomposes shapes into both surface and internal structures. Extensive experiments show that PartSAM outperforms state-of-the-art methods by large margins across multiple benchmarks, marking a decisive step toward foundation models for 3D part understanding. Project page: https://czvvd.github.io/PartSAMPage/.

# 1 INTRODUCTION

Segmenting 3D objects into their constituent parts is a fundamental problem in computer vision and graphics. A reliable solution would benefit a wide range of downstream applications, including 3D asset creation, AR/VR content editing, and robotic manipulation. The key requirements of such a model are generalization beyond fixed taxonomies, flexible interaction through user guidance, and robustness in open-world scenarios where novel categories and diverse part definitions are common.

Conventional approaches (Qi et al., 2017; Mo et al., 2019; Wang et al., 2019a; Zhao et al., 2021) have made progress by training networks on 3D datasets with predefined part taxonomies. However, these datasets are small and closed-world in nature: for example, chairs may only be annotated with seat, back, and legs, while cars are annotated with wheels and doors. Models trained under such assumptions cannot generalize to unseen categories or alternative definitions of part granularity. As a result, they perform well within benchmarks but fall short when deployed in real-world scenarios with open-world queries.

To address these limitations, subsequent works (Liu et al., 2023; Zhu et al., 2023; Liu et al., 2024a; 2025; Ma et al., 2024) transfer knowledge from extensively trained 2D foundation models. For example, SAMPart3D (Yang et al., 2024b) lifts multi-view segmentation results of SAM (Kirillov et al., 2023) to 3D space, but this process requires time-consuming per-shape optimization. More recent methods (Ma et al., 2024; Liu et al., 2025; Zhou et al., 2025) adopt a feed-forward paradigm, training 3D networks with supervision from SAM's 2D masks and directly inferring segmentation results.

For instance, PartField (Liu et al., 2025) achieves impressive performance by training a 3D feature field through contrastive learning, with segmentation results obtained by clustering in the feature space. Despite these advancements, existing methods still lag significantly behind their 2D counterpart, SAM, due to two key limitations: (1) Clustering-based segmentation approaches (Yang et al., 2024b; Liu et al., 2025) lack the user-centered controllability inherent to SAM, often resulting in fragmented parts without a carefully tuned cluster number. (2) These methods rely heavily on supervision from SAM's multi-view 2D segmentation, which restricts their capabilities to the object's surface, thus limiting their ability to achieve comprehensive 3D geometric understanding and to capture interior structures (see Figure 2).

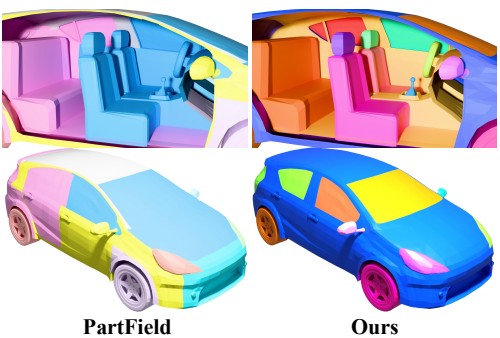

**PartField**      **Ours**

Figure 2: The SOTA method PartField (Liu et al., 2025) fails to segment the interior structure of 3D shapes.

In this paper, we propose PartSAM, a promptable part segmentation model trained natively on large-scale 3D data, which addresses the above challenges in three aspects. First, our network adopts a segmentation paradigm inspired by SAM, featuring a prompt-guided encoder-decoder architecture. Such a SAM-like decoder facilitates more accurate grouping of semantically consistent parts than clustering-based segmentation, as the prompt-guided training provides a more precise supervisory signal. Moreover, as the segmentation is explicitly controlled by prompts, it supports flexible interactive or automatic segmentation during inference.

Second, we propose an effective encoder that can scale to large 3D datasets. Specifically, we encode shapes into a triplane-based feature field (Chan et al., 2022) using a dual-branch transformer (Vaswani et al., 2017). To better leverage existing priors, we leverage one learnable branch scale to large 3D datasets, while the other frozen path preserves 2D priors of SAM learned through contrastive learning (Liu et al., 2025). Additionally, by incorporating input attributes beyond just coordinates, the model enhances the representation of local shape details.

Third, we propose a scalable pipeline for curating a large-scale part segmentation dataset. We extract part supervision from extensive 3D assets (Deitke et al., 2023b;a) by leveraging artist-annotated scene graphs and connected components. To ensure sufficient diversity in the supervision, we design a model-in-the-loop strategy that iteratively refines and scales annotations, ultimately producing over five million native 3D shape–part pairs.

Compared with existing methods (Yang et al., 2024b; Liu et al., 2025; Zhou et al., 2025; Ma et al., 2024), PartSAM is the first to simultaneously achieve flexible controllability, feed-forward inference, and scalable performance within a native 3D framework. The synergy of a scalable architecture and large-scale 3D training data equips PartSAM with generalizable interactive capabilities. When given only a single prompt point, PartSAM surpasses Point-SAM (Zhou et al., 2025) by over 90% in open-world settings. Building upon the impressive single-prompt segmentation quality, we further propose a "Segment Every Part" mode to automatically segment entire shapes. Experiments show that PartSAM significantly outperforms state-of-the-art methods in this task. The main contributions of this work can be summarized as follows.

- We introduce PartSAM, the first scalable feed-forward promptable 3D part segmentation model trained with native 3D supervisions, enabling flexible open-world segmentation at inference.
- We design an effective dual-branch encoder that represents 3D shapes as robust part-aware feature fields, enabling effective scaling to native 3D supervision, while simultaneously retaining the powerful 2D priors derived from SAM.
- We propose a model-in-the-loop annotation pipeline to mine part supervision from large-scale 3D assets, scaling the training data to millions of shape-part pairs.
- PartSAM achieves state-of-the-art performance on multiple open-world part segmentation benchmarks and demonstrates broad applicability, providing a strong foundation for future research in 3D part understanding.

## 2 RELATED WORK

### 2.1 CLOSED-WORLD 3D PART SEGMENTATION

Early learning-based methods (Qi et al., 2017; Thomas et al., 2019; Wang et al., 2019b; Zhang & Wonka, 2021) typically formulate part segmentation as point-level semantic or instance classification on 3D shapes. These approaches are commonly trained on ShapeNet-Part (Yi et al., 2016) and PartNet (Mo et al., 2019). The limited diversity of these datasets in terms of object and part categories, however, constrains their generalization to unseen data.

### 2.2 LIFTING 2D FOUNDATION MODELS FOR 3D PART SEGMENTATION

To address the limitations of existing small datasets and enable open-world part segmentation, a growing body of work leverages the strong priors embedded in 2D foundation models (Radford et al., 2021; Li et al., 2022; Kirillov et al., 2023; Ravi et al.; Oquab et al., 2024). One research direction focuses on text-driven approaches (Abdelreheem et al., 2023; Liu et al., 2023; Zhu et al., 2023; Garosi et al., 2025), which query target segmentation based on the part name in the feature space of vision–language models (Radford et al., 2021; Li et al., 2022).

Another prominent line of work explores transferring knowledge from SAM (Kirillov et al., 2023; Ravi et al.). Some methods directly project SAM's multi-view segmentation masks into 3D space by exploiting relationships across views (Liu et al., 2024a; Zhong et al., 2024; Tang et al., 2024). For example, SAMesh (Tang et al., 2024) employs a community detection algorithm to merge multi-view predictions into coherent 3D segments. Other approaches instead distill SAM in the feature space (Yang et al., 2024b; Lang et al., 2024). SAMPart3D (Yang et al., 2024b) introduces a scale-conditioned MLP to handle ambiguity in SAM-generated masks, while iSeg (Lang et al., 2024) adopts a two-stage pipeline that distills SAM's features for 3D interactive segmentation.

Despite the progress, these lifting-based methods still face inherent challenges. In practice, they often require computationally expensive post-processing for each shape during inference, which restricts their scalability in downstream applications.

### 2.3 FEED-FORWARD MODELS FOR 3D SEGMENTATION

Motivated by the scaling laws observed in modern large models, recent efforts have aimed to train feed-forward 3D segmentation models that directly predict open-world results at inference time. Prior work on scene segmentation (Takmaz et al., 2023; Wang et al., 2025; Jiang et al., 2024; Peng et al., 2023; Yang et al., 2024a) mainly focuses on aligning 3D representations with text embeddings in a vision–language feature space through 2D–3D aggregation. While these methods demonstrate strong performance on scene-level tasks, extending them to object part segmentation is considerably more challenging, as it requires fine-grained reasoning about part-level geometry.

For part segmentation specifically, Find3D (Ma et al., 2024) adopts a training strategy similar to these scene-level methods. To address the data scarcity problem, it uses SAM and Gemini to generate 2D part annotations with associated textual labels. PartField (Liu et al., 2025) instead represents shapes as continuous feature fields and trains a transformer-based feed-forward network with an ambiguity-agnostic contrastive loss. Despite its impressive performance, the feature space clustering of PartField relies extremely on the connectivity of the input mesh to get a high-quality segmentation. For generated meshes (Lai et al., 2025; Xiang et al., 2025; Zhang et al., 2024; Liu et al., 2024b; Long et al., 2024), where such connectivity is absent, performance drops significantly (see Fig. 8). Zhou et al. (2025) further propose Point-SAM, a 3D analogue of SAM trained on SAM-annotated multi-view renderings of ShapeNet, but the limited encoder scalability and training data diversity restrict its generalization to open-world settings. Consequently, it does not fully unleash the potential of the SAM architecture for part segmentation.

More broadly, all these existing approaches heavily rely on SAM-generated 2D masks for data construction, which fundamentally constrains their capacity to capture intrinsic 3D geometric structure and predict meaningful interior parts. Distinct from these methods, we introduce a native 3D foundation model for part segmentation, trained on large-scale shape–part pairs to enable scalable, geometry-aware, and controllable understanding beyond 2D supervision.

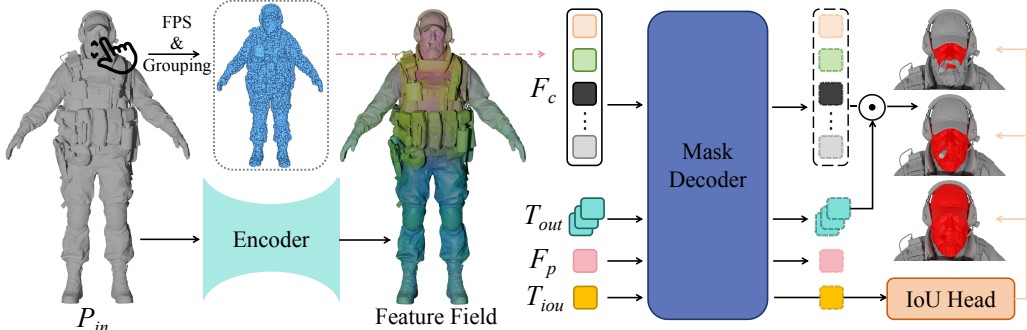

Figure 3: Overview of the PartSAM model. The input shape $P_{in}$ is first encoded into a continuous feature field. Point patches sampled from $P_{in}$ query this field to obtain input embeddings $F_c$, while prompt points are mapped into prompt embeddings $F_p$. Both $F_c$ and $F_p$ are fed into the mask decoder, where the learnable output token $T_{out}$ generates multiple segmentation masks. An additional IoU token $T_{iou}$ is used by the IoU head to estimate the quality of each mask.

## 3 METHOD

An overview of PartSAM is shown in Figure 3. Our goal is to develop a scalable framework for segmenting arbitrary 3D parts, analogous to SAM (Kirillov et al., 2023) in the image domain. The architecture consists of two main components: an input encoder that transforms 3D shapes into structured feature embeddings, and a prompt-guided mask decoder that predicts segmentation masks conditioned on user prompts. We primarily consider 3D click points as prompts, including both positive and negative ones. Formally, given a 3D shape represented as a point cloud $P_{in} \subseteq \mathbb{R}^{N \times d_{in}}$, where each point may include 3D coordinates, surface normals, and optional RGB color (i.e., $d_{in} = 9$), and a set of user prompts $P_{prompt} \subseteq \mathbb{R}^{N_p \times 3}$, PartSAM first transform the input shape into feature embeddings $F_c$ through the input encoder. These are combined with prompt embeddings $F_p$ in the mask decoder to generate a binary segmentation mask $M_{out}$. To fully realize this design at scale, we

further introduce a data curation pipeline that mines diverse and fine-grained 3D part annotations from large-scale assets, enabling PartSAM to generalize across open-world scenarios. Additional details are provided in Appendix A.1.

### 3.1 INPUT ENCODER

The encoder, a key component for scalable training, extracts feature embeddings $F_c$ from $P_{in}$ for the mask decoder. Unlike prior approaches (Zhou et al., 2025) that rely solely on point cloud networks, our approach encodes parts within a continuous triplane (Chan et al., 2022) feature field.

As illustrated in Figure 4, we implement the encoder as a dual-branch network to effectively leverage priors from different modalities. In each branch, the input points are first transformed into three 2D planes using PVCNN (Liu et al., 2019), followed by a projection operation. These axis-aligned planes are then processed by a transformer (Vaswani et al., 2017), allowing them to query feature vectors for any given 3D coordinate. During training, we initialize each branch with pre-trained weights from PartField (Liu et al., 2025), keeping one branch frozen and the other one learnable. The frozen branch incorporates rich 2D knowledge distilled from SAM (Ravi et al.), learned via contrastive learning (Liu et al., 2025), while the learnable branch adapts and learns new representations of native 3D parts from our training data. Since our SAM-like architecture can not be directly trained with incomplete 2D masks, this dual-branch design enables effective scaling to native 3D supervision, while simultaneously retaining the powerful 2D priors derived from SAM. Moreover, the learnable branch accepts additional input attributes beyond coordinates (i.e., normal and RGB) through a zero convolution layer (Zhang et al., 2023),

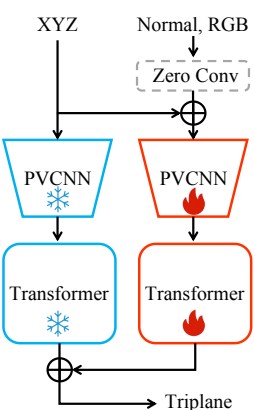

Figure 4: Architecture of our dual-branch encoder. Each branch is initialized with pre-trained weights of Liu et al. (2025).

further enhancing the representation of shape details. Finally, the outputs of the two branches are summed to produce a continuous feature field.

To obtain tokens for the mask decoder, we adopt the sampling-and-grouping strategy of Qi et al. (2017), where $N_c$ centers are selected via farthest point sampling (FPS), local patches are formed around each center, and features sampled from the triplane are aggregated with a shared MLP:

$$F_c = \text{MLP}\Big(\{\phi(p) \mid p \in \mathcal{N}(\text{FPS}(P_{in}, N_c))\}\Big) \in \mathbb{R}^{N_c \times C}, \tag{1}$$

where $\phi(\cdot)$ denotes feature sampling from the triplane and $\mathcal{N}(\cdot)$ means KNN-based local patches.

### 3.2 PROMPT-GUIDED MASK DECODER

Unlike the clustering-based methods (Yang et al., 2024b; Liu et al., 2025), the prompt-guided decoder directly produces segmentation masks explicitly controlled by user prompts, enabling flexible interactive or automatic segmentation during inference. Specifically, it maps user prompt points $P_{prompt}$ and input embeddings $F_c$ to a binary segmentation mask $M_{out}$. Prompt points are first encoded into embeddings $F_p$ by sampling features from the continuous feature field and combining them with position embeddings. For multi-round interactions, mask logits from previous rounds are incorporated as additional prompts and directly added to $F_c$ to refine the predictions.

For mask decoding, two special tokens are introduced: an output token $T_{out}$ that generates segmentation masks, and an IoU token $T_{iou}$ that estimates mask quality. These tokens, together with the prompt embeddings $F_p$, are concatenated and attend bidirectionally to the patch embeddings $F_c$ using a two-way transformer (Kirillov et al., 2023):

$$F'_c = \text{CrossAttn}\big(F_c \leftrightarrow [F_p; T_{out}; T_{iou}]\big) \tag{2}$$

The refined embeddings $F'_c$ are upsampled to the input resolution using distance-based interpolation. Mask logits are subsequently computed through a point-wise dot product between the upsampled embeddings and the refined output token $T'_{out}$. Per-point foreground probabilities are then obtained by applying a sigmoid function, and the final binary mask $M_{out}$ is generated by thresholding these probabilities.

To handle the inherent ambiguity of 3D part boundaries, the decoder follows the parallel decoding strategy of SAM. When a single prompt is given, multiple output tokens (three in our implementation) generate diverse candidate masks. In parallel, an additional IoU token is trained to estimate the overlap between each predicted mask and the ground truth, providing a confidence score that guides the selection of the final output.

### 3.3 TRAINING STRATEGY

Following SAM (Kirillov et al., 2023), during training, we simulate interactive segmentation by first sampling a prompt from the center of the ground-truth mask and then iteratively sampling subsequent prompts from prediction error regions for 9 iterations. The IoU prediction is supervised with the MSE loss. The segmentation masks are supervised with a combination of focal loss (Lin et al., 2017) and dice loss (Milletari et al., 2016). For each ground-truth mask, the triplet contrastive loss in Liu et al. (2025) is also used to strengthen the representation ability of the encoder.

$$\mathcal{L} = \mathcal{L}_{focal}(M_{out}, M_{gt}) + \alpha \mathcal{L}_{dice}(M_{out}, M_{gt}) + \mathcal{L}_{IoU} + \lambda \mathcal{L}_{triplet}, \tag{3}$$

where $M_{gt}$ is the ground-truth mask and $\alpha$ and $\lambda$ are weighting coefficients for the loss terms.

### 3.4 AUTOMATIC SEGMENTATION

Similar to SAM (Kirillov et al., 2023), PartSAM demonstrates an emergent ability to autonomously segment complete shapes after training. To facilitate this, we propose a novel "Segment Every Part" pipeline for utilizing our decoder to automatically segment every meaningful part for a shape. Concretely, we first sample $N_f$ points from each shape using FPS, treating each sampled point as an independent prompt. This yields $3N_f$ candidate masks together with their predicted IoU scores. The candidates are then refined in two stages: (1) discarding masks with low predicted IoU, and (2) applying Non-Maximum Suppression (NMS) based on point-level IoU to eliminate redundant masks, with the predicted IoU serving as the confidence score. Finally, mesh faces are assigned labels according to the point-level predictions sampled on them. During this process, the threshold value of NMS $T$ is set as a hyperparameter to adjust the granularity of automatic segmentation.

### 3.5 DATA CURATION

Training a scalable and generalizable 3D segmentation model requires large-scale, high-quality part annotations. To this end, we construct such supervision through two complementary stages:

**Integrating existing part labels.** We primarily curate part annotations from Objaverse (Deitke et al., 2023b) and other licensed datasets. For existing part segmentation datasets like PartNet (Mo et al., 2019) and ABO (Collins et al., 2022), we directly use the provided ground-truth labels. For artist-created assets, like those in Objaverse, GLTF scene graphs can serve as natural supervision when available, while assets with a single geometry node are decomposed into connected components. Then, to improve annotation reliability, we discard shapes with fewer than three or more than fifty parts and filter out components that are extremely small or excessively large. This process yields approximately 180k shapes with 2 million parts.

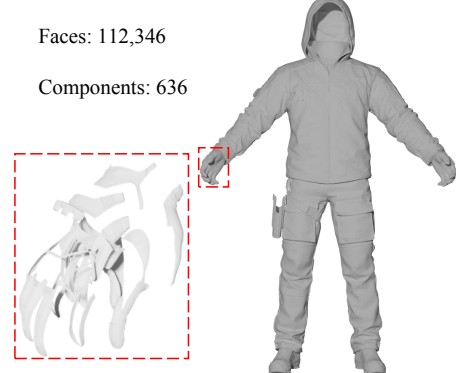

Faces: 112,346

Components: 636

Figure 5: Example of an artist-created mesh with over 600 connected components. The large number of fragmented pieces makes it difficult to obtain semantically meaningful parts, and such assets are excluded from direct supervision.

Most highly fragmented assets are excluded during this filtering stage, since their connected components fail to form semantically meaningful parts (see Figure 5). While these cues cannot be directly utilized for training, they motivate our next step: we first pretrain PartSAM on the data from the first stage, and then leverage the pretrained model in a model-in-the-loop pipeline to annotate part labels for these over-fragmented structures.

**Model-in-the-loop annotation.** Although PartField (Liu et al., 2025) suffers from limited controllability due to its clustering-based design, it can still produce sharp masks for certain parts when mesh connectivity is well defined. We exploit this property by treating PartField outputs as candidate labels and leveraging the interactive capability of PartSAM to filter out noisy segmentations. Concretely, we first pretrain PartSAM on the curated dataset, which, though less diverse, endows the model with basic interactive part segmentation ability. For each over-fragmented shape discarded in the first stage, PartField generates multi-scale masks by setting the clustering number to 10, 20, and 30. These masks are used as pseudo labels and passed to PartSAM, which performs 10 rounds of interactive segmentation simulation (Sec. 3.3). At each step, we compute the IoU between the PartSAM prediction and the PartField pseudo label, denoted IoU@$i$ for the $i$-th iteration. A mask is considered valid if it satisfies either IoU@1 $> 60$ or IoU@10 $> 90$. The intuition is that reliable part masks should either be interactively segmented immediately with even a single prompt, or be progressively refined through extended interactions. Meanwhile, we only regard a shape as valid if it contains more than 5 valid masks. As illustrated in Figure 15, this model-in-the-loop process yields robust annotations. Through this pipeline, we expand the training corpus to $500k$ shapes and more than 5 million parts, substantially improving both scale and quality.

# 4 EXPERIMENT

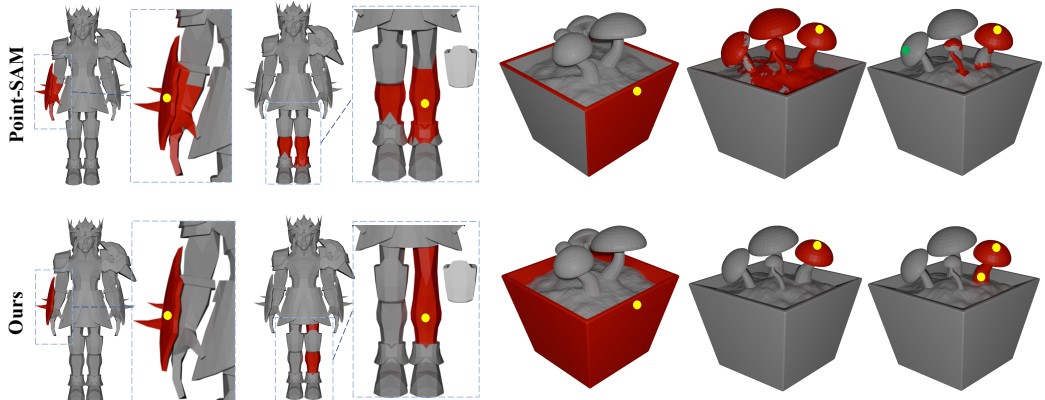

Figure 6: Qualitative comparison with Point-SAM (Zhou et al., 2025) on interactive part segmentation. Predicted segmentation masks are shown in red. Yellow and green points denote positive and negative prompt points, respectively. Compared to Point-SAM, PartSAM produces more complete and semantically consistent parts, even with minimal prompts.

Table 1: Quantitative comparison of interactive segmentation on PartObjaverse-Tiny (Yang et al., 2024b) and PartNetE (Liu et al., 2023). The **best** scores are emphasized in bold. IoU@$i$ denotes mean IoU value with $i$ prompt points. We report the mean IoU on instance-level labels.

| Dataset | Method | IoU@1 | IoU@3 | IoU@5 | IoU@7 | IoU@10 |
|---------|--------|-------|-------|-------|-------|--------|
| PartObjaverse-Tiny | Point-SAM | 29.4 | 58.6 | 68.7 | 71.8 | 73.9 |
| | Ours | **56.1** | **78.3** | **84.1** | **86.2** | **87.6** |
| PartNetE | Point-SAM | 35.9 | 68.0 | 75.1 | 77.6 | 79.2 |
| | Ours | **59.5** | **79.3** | **86.5** | **88.3** | **89.9** |

## 4.1 COMPARISON OF INTERACTIVE PART SEGMENTATION

We compare PartSAM with Point-SAM (Zhou et al., 2025) on the interactive part segmentation task using the PartObjaverse-Tiny (Yang et al., 2024b) and PartNet-E (Liu et al., 2023) datasets. Following the experimental protocol of Zhou et al. (2025), the first prompt point for each ground-truth mask is sampled from its central region. Subsequent prompt points are iteratively selected from the error regions between the predicted mask and the ground truth.

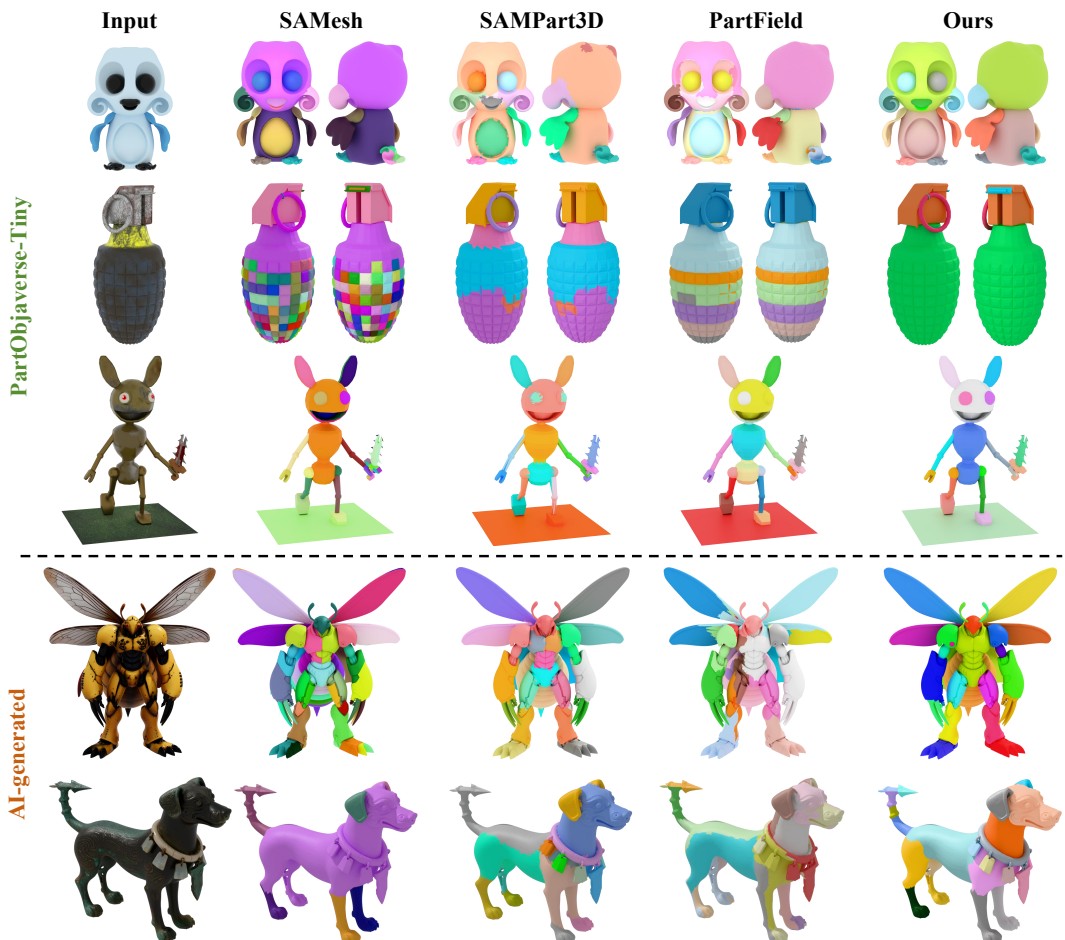

Figure 7: Qualitative comparison of class-agnostic part segmentation with baselines (Tang et al., 2024; Yang et al., 2024b; Liu et al., 2025) on PartObjaverse-Tiny (Yang et al., 2024b) and AI-generated 3D models (Lai et al., 2025). Each segmented part is visualized with a distinct color.

The quantitative results in Table 1 demonstrate PartSAM's superiority over Point-SAM across varying numbers of prompt points. In particular, with only a single prompt, PartSAM achieves a $91\%$ relative improvement in IoU, demonstrating its ability to accurately delineate parts from just one click. This advantage is further supported by the qualitative results in Figure 6. PartSAM reliably produces precise and semantically meaningful parts even from a single point, whereas Point-SAM often fails due to the limited scalability of the network architecture, thus lacking an explicit notion of arbitrary 3D parts in the open world setting. Moreover, Point-SAM's dependence on 2D multi-view supervision from SAM hampers its ability to capture internal 3D structures. For instance, in the second column of Figure 6, PartSAM successfully segments the entire leg, including the occluded knee, while Point-SAM misses it due to its incomplete understanding of 3D geometry.

## 4.2 COMPARISON OF CLASS-AGNOSTIC PART SEGMENTATION

As introduced in Section 3.4, PartSAM supports automatic shape decomposition through its "segment every part" mode. We compare PartSAM against five state-of-the-art methods (Liu et al., 2023; Ma et al., 2024; Yang et al., 2024b; Tang et al., 2024; Liu et al., 2025) on this task. We use the PartObjaverse-Tiny (Yang et al., 2024b) and PartNet-E (Liu et al., 2023) datasets for quantitative evaluation and additionally use AI-generated meshes produced by Hunyuan3D (Lai et al., 2025) for qualitative comparison. We follow the experimental protocol from Liu et al. (2025) with a key modification to prevent potential label leakage from mesh connectivity. Specifically, since artist-created meshes in PartObjaverse-Tiny contain strong connectivity cues that align with ground-

PartField                 Ours

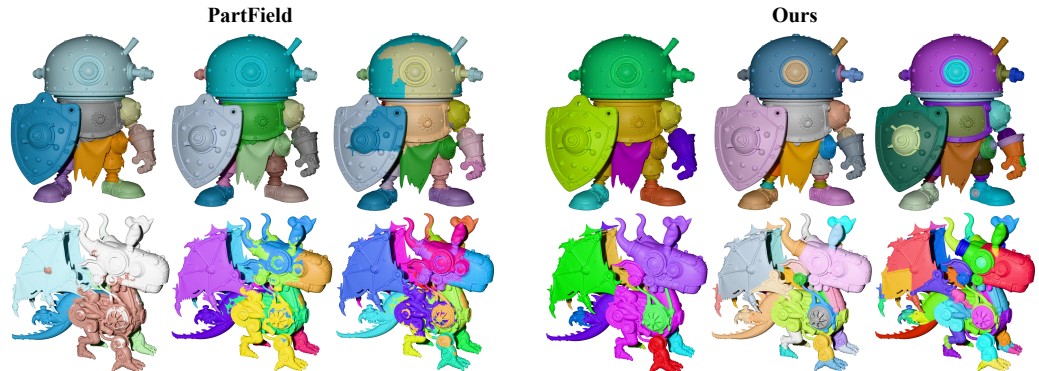

Figure 8: Qualitative comparison of hierarchical part segmentation with PartField (Liu et al., 2025) on AI-generated 3D models (Lai et al., 2025). Each segmented part is represented as a distinct color.

truth part labels (see Appendix A.2.5), we disable this connectivity information when evaluating PartField (Liu et al., 2025) and instead apply K-Means clustering directly on mesh faces. In contrast, SAMesh (Tang et al., 2024), which fundamentally depends on mesh connectivity through community detection, is evaluated in its original setting. Notably, for baselines that produce multi-scale part segmentations, we evaluate performance by computing the IoU for all candidate masks and reporting the best score. For our PartSAM, multi-scale segmentations are obtained by varying the NMS threshold value $T$ at 0.1, 0.3, 0.5, and 0.7, respectively.

Table 2: Quantitative comparison of automatic segmentation on PartObjaverse-Tiny (Yang et al., 2024b) and PartNetE (Liu et al., 2023). * denotes that PartField is evaluated with K-Means clustering without mesh connectivity information. We report the mean IoU on instance-level labels.

| Dataset | PartSLIP | Find3D | SAMPart3D | SAMesh | PartField* | Ours |
|---|---|---|---|---|---|---|
| PartObjaverse-Tiny | 31.5 | 21.3 | 53.5 | 56.9 | 51.5 | **69.5** |
| PartNetE | 34.9 | 21.7 | 56.2 | 26.7 | 59.1 | **72.4** |

Results in Table 2 demonstrate that PartSAM consistently surpasses all competing methods, achieving over 20% IoU improvement over the second-best approach across both datasets.

Qualitative comparisons in Figure 7 further reveal that PartSAM produces robust part segmentations for diverse 3D shapes. SAMesh also exhibits clear boundaries but often generates over-segmented outputs, as its direct projection and merging of 2D SAM masks into 3D space fails to capture underlying geometric structures. SAMPart3D and PartField, which rely on feature clustering, encounter difficulties on complex and detailed structures. Their clustering process frequently yields fragmented or semantically meaningless parts, for example, incorrectly partitioning the body of a grenade into several disjoint segments.

We also conduct a comparison with PartField in the context of hierarchical multi-scale segmentation on AI-generated meshes (Figure 8). PartSAM delivers finer-grained results with sharper boundaries, highlighting its capability to produce high-quality segmentation across scales.

In Figure 9, we further compare PartSAM with baselines on diverse AI-generated meshes with interior or occluded parts. SAMesh benefits from inheriting 2D SAM's billion-scale mask priors, enabling it to capture extremely fine surface details and produce sharp boundaries (e.g., the car exterior). However, this 2D-driven granularity may also over-fragment large but semantically coherent regions (handbag), and, more importantly, it struggles when parts are occluded in most rendered views—for example, SAMesh cannot reliably recover the objects inside the handbag, the seats and steering wheel inside the car, or the structures hidden beneath the robot's cloak. In contrast, PartSAM operates directly on native 3D geometry and learns multiscale, semantically meaningful decompositions, producing coherent part groupings for both visible and partially hidden structures.

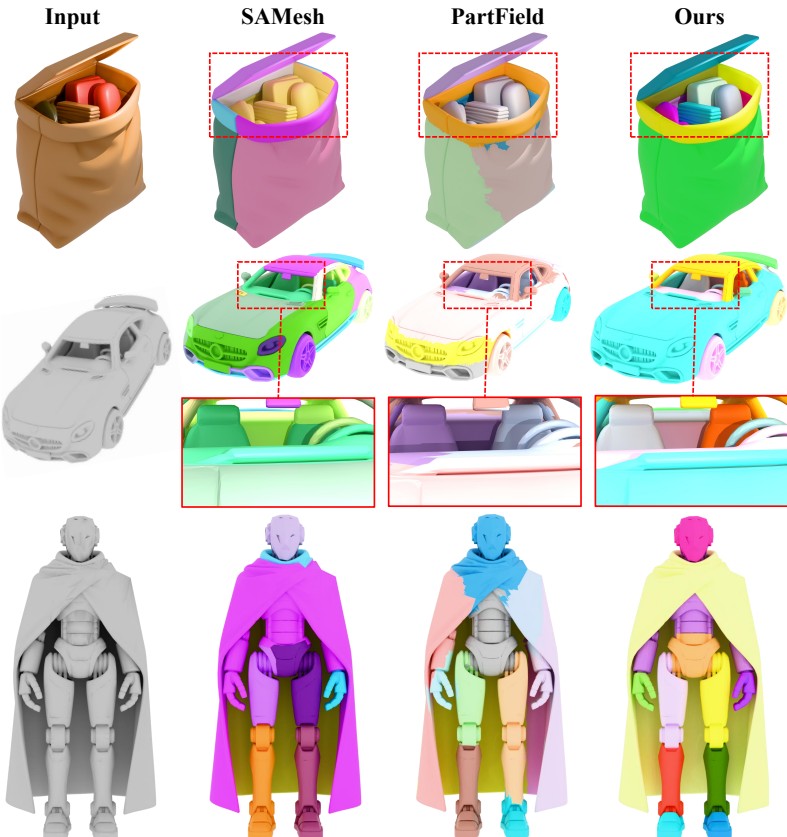

Figure 9: Qualitative comparison of class-agnostic part segmentation with baselines (Liu et al., 2025; Tang et al., 2024) on AI-generated 3D models (Lai et al., 2025) with interior structures and occluded parts. Each segmented part is represented as a distinct color.

Overall, these results highlight the superiority of PartSAM over existing methods: By replacing heuristic clustering with an automatic promptable framework and leveraging large-scale 3D training, it establishes a more controllable and generalizable model with 3D-aware part understanding.

## 5 CONCLUSION AND DISCUSSIONS

In this work, we introduced PartSAM, a scalable promptable model that performs part segmentation natively in 3D. Unlike prior approaches that rely on transferring knowledge from 2D foundation models, PartSAM is trained directly on millions of 3D shape–part pairs, enabling faithful understanding of intrinsic 3D geometry. Our design combines a triplane-based encoder with a promptable encoder–decoder architecture, supported by a model-in-the-loop pipeline that supplies diverse annotations at scale. This synergy between architecture and data drives state-of-the-art performance on both interactive and automatic segmentation tasks. These findings suggest that scaling with native 3D data opens new avenues for advancing part-level understanding, and we believe PartSAM lays a foundation for future exploration of richer, more generalizable 3D perception models.

**Discussions.** Please refer to the Appendix for comprehensive ablation studies and additional discussions, including applications, complexity analysis, and limitations.

## 6 ACKNOWLEDGMENTS

This work was supported by the National Natural Science Foundation of China (No. T2322012, No. 62572240). It was also supported by the donation from Kerry Group Limited in "30-for-30" Talent Acquisition Campaign of HKUST.

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

# A  APPENDIX

## A.1  ADDITIONAL DETAILS

### A.1.1  DETAILS OF THE SEGMENTATION PIPELINE

**Overall.** Given an input point cloud $P_{\text{in}} \in \mathbb{R}^{N \times d}$ and a set of prompt points $P_{\text{prompt}} \in \mathbb{R}^{N_p \times 3}$, PartSAM predicts a binary part mask $M_{\text{out}}$.

**Encoder.** The encoder contains two triplane branches. Each of them follows a similar architecture to Liu et al. (2025). The learnable branch differs only in its input layer, where six additional feature channels (representing XYZ coordinates and normals) are concatenated to the coordinates. Within each branch, a PVCNN (Liu et al., 2019) operating at a voxel resolution of $32^3$ first extracts per-point features, which are orthogonally projected onto three axis-aligned planes to form the initial triplane field $F_{\text{plane}} \in \mathbb{R}^{3 \times H \times W \times C^{\text{init}}}$. The planes are then downsampled by a factor of $r$ via a two-layer CNN and reshaped into a sequence of size $(3HW/r^2) \times C^{\text{trans}}$, which is subsequently processed by a transformer. The transformer outputs are upsampled through a transposed convolution layer, reshaped back into three planes, and summed, yielding a triplane representation of size $3 \times H \times W \times C$. Point-wise features are sampled from the resulting planes according to their coordinates. A sampling-and-grouping operation (Qi et al., 2017) further aggregates these features to produce the input embeddings $F_c \in \mathbb{R}^{N_c \times C}$.

**Decoder.** For a single prompt, we use three output tokens and one IoU token, resulting in three predicted masks. For multiple prompts, we follow SAM (Kirillov et al., 2023) by adding an additional output token that predicts a single aggregated mask, which is ignored in the single-prompt case and used exclusively in the multi-prompt setting. Thus, in both settings, the decoder always receives five special tokens (three output tokens, one IoU token, and one auxiliary output token). Prompt points are encoded by sampling the triplane feature field at their coordinates and adding positional and prompt-type (positive or negative) embeddings, producing $F_p \in \mathbb{R}^{N_p \times C'}$. We project the input embeddings $F_c$ to the same dimension $C'$ via a linear layer, yielding $F_c \in \mathbb{R}^{N_c \times C'}$. Output tokens and the IoU token are concatenated with $F_p$ to form a unified token set with shape of $(N_p + 5) \times C'$, which is processed jointly with $F_c$. The two sets interact through a four-layer two-way transformer (similar to Kirillov et al. (2023)), where each layer applies self-attention within tokens, followed by cross-attention from tokens to embeddings and another cross-attention from embeddings back to tokens. This produces refined output token $T'_{\text{out}}$, refined IoU token $T'_{\text{iou}}$, and updated input embeddings $F'_c$. The refined input embeddings $F'_c$ are upsampled to per-point resolution via distance-based interpolation followed by an MLP, yielding a tensor of shape $N \times C'$. Then, each refined $C'$-dimensional output token predicts mask logits through a dot product with the upsampled embeddings, which is passed through a sigmoid and then thresholded to obtain a binary mask.

**IoU head.** When a single prompt is given, the refined IoU token $T'_{\text{iou}} \in \mathbb{R}^C$ is passed through a three-layer MLP to predict mask-wise IoU scores. The mask with the highest score is then selected as the final output $M_{\text{out}}$.

**Hyperparameters.** In all experiments, we follow the setup of Liu et al. (2025) and uniformly sample the input point cloud to $N_p = 100,000$ points. For shapes without texture, we assign all points a default gray color. Our encoder is configured with a triplane resolution of $H = W = 512$, $C^{init} = 64$, $C^{trans} = 1024$, $C = 448$, and 6 transformer layers per branch. The number of point patches $N_c$ is set to $2,000$. The feature dimension of two-way transformer layers $C'$ is set to $256$.

### A.1.2  MECHANISM OF INCORPORATING PREVIOUS POINT PROMPT

In the first round of interaction, we input the first prompt point and obtain the initial mask result. In subsequent rounds, we concat the last predicted mask logits with coordinates and adopt the same sampling-and-grouping strategy as in Equation (1) to extract dense mask embeddings $F_p \subseteq \mathbb{R}^{N_c \times C}$. These dense embeddings are directly added to the input embeddings $F_c$, allowing the model to incorporate information from previous prompts. For the new prompt point, we distinguish positive and negative prompts by adding different learnable prompt-type embeddings to $F_p$.

### A.1.3 DATA CURATION DETAILS

Our data curation follows a fully automated two-stage pipeline designed to construct large-scale, high-quality 3D part annotations. In the first stage, several safeguards are applied to filter out low-quality data from Objaverse(-XL). We remove scanned objects that generally exhibit poor geometric fidelity and are unsuitable for part annotation. We further discard parts that are excessively small or large based on their foreground point statistics, and exclude objects containing fewer than 3 or more than 50 parts to maintain reasonable segmentation granularity. In the second stage, we focus exclusively on high-poly, artist-designed meshes that provide rich geometric structure and numerous connected components. Artist-defined connectivity information is incorporated into PartField's clustering process to produce clean, well-separated segments. A model-in-the-loop procedure then imposes two additional safeguards: (i) a part-level filter that accepts only segments meeting the interactive IoU criterion, and (ii) a shape-level filter that retains only shapes with more than five valid parts. Only shape–part pairs satisfying all conditions are preserved, ensuring that clustered masks lacking clear semantic meaning (e.g., wheels merged into the car body) are consistently removed.

To ensure that the pipeline scales to millions of shape–part pairs, the whole data curation process is designed to be fully automated. Although a very small number of imperfect cases may remain due to the absence of human verification, their proportion is negligible and does not affect model behavior. As evidenced by the scaling curves in Figure 11, model performance continues to improve as dataset size increases, demonstrating both the scalability of the model and the overall high quality of the curated training data.

### A.1.4 TRAINING DETAILS

During training, we simulate the interactive segmentation process as follows: For each ground-truth foreground mask, we first sample the prompt point from the foreground region, selecting points whose distance to the background falls within the bottom third of the total distance range. After obtaining the predicted masks, we iteratively sample subsequent prompt points by selecting the furthest points within the error regions (i.e., the areas where the predicted and ground-truth masks differ). This process is repeated 9 times per mask. We set $\alpha = 2$ and $\lambda = 0.5$. A comprehensive data augmentation strategy is employed to improve model robustness by introducing controlled variations in geometry, appearance, and scale. The augmentation pipeline includes random rotations, scaling, flipping, and center shifting of the point cloud, as well as chromatic transformations such as auto contrast, translation, and jitter. Overall, PartSAM is trained using the AdamW optimizer for $250k$ iterations with a batch size of $4$ on 8 NVIDIA H20 GPUs. The learning rate is initialized at $5 \times 10^{-4}$ and decayed by a factor of $0.7$ every $50k$ iterations.

### A.2 ABLATION STUDY AND DISCUSSIONS

We ablate PartSAM by removing or modifying its key components and evaluate both interactive and automatic segmentation, as reported in Table 3.

Table 3: Ablation study on PartObjaverse-Tiny (Yang et al., 2024b). We report the mean IoU on instance-level labels for both the interactive and automatic segmentation tasks.

| Method | IoU@1 | IoU@3 | IoU@5 | IoU@7 | Automatic |
|---|---|---|---|---|---|
| **Model Variants** | | | | | |
| Point-SAM | 38.9 | 62.9 | 71.5 | 76.8 | 48.6 |
| w/o pre-trained weights | 48.3 | 73.6 | 79.1 | 80.5 | 60.5 |
| Frozen PartField | 42.5 | 69.4 | 73.6 | 78.3 | 54.3 |
| Learnable PartField | 50.8 | 73.9 | 79.9 | 83.2 | 61.8 |
| **Training Data Variants** | | | | | |
| PartNet | 33.7 | 59.1 | 67.3 | 70.5 | 40.2 |
| w/o Model-in-the-loop Annotation | 49.0 | 72.3 | 78.7 | 81.1 | 62.6 |
| **Ours** | **56.1** | **79.0** | **84.7** | **86.9** | **68.5** |

### A.2.1 ABLATION OF MODEL ARCHITECTURE

To assess the impact of the model architecture, we compare three variants of PartSAM trained under identical settings.

**Comparison with Point-SAM.** To comprehensively compare with the existing promptable segmentation model Point-SAM (Zhou et al., 2025), we train it using our curated dataset and report the results in Table 3. The significant performance drops on both the interactive and automatic segmentation tasks indicate that Point-SAM fails to scale on our native 3D data. Compared with Point-SAM, our method offers advantages in both network architecture and training scheme for the part segmentation task. Regarding architecture, Point-SAM uses a transformer encoder operating on unstructured point clouds, whereas our method performs transformer modeling on a more advanced triplane feature field. The dual-branch encoder also brings in 2D SAM priors, which improve generalization and stabilize training on large-scale 3D data. We also benefit from the training scheme: the triplet contrastive loss strengthens part-aware feature learning, and its formulation is naturally robust to the scale and granularity ambiguity of 3D parts. Together, the continuous feature representation and the contrastive learning scheme give our framework substantially stronger scalability.

**Analysis of Pre-trained Encoder.** Then, we evaluate a variant that trains PartSAM entirely from scratch, without using any initialization. This variant performs worse in terms of all tasks, indicating that the SAM-derived priors provide useful part-aware cues while not introducing lifting-related artifacts. Next, we employ the pre-trained PartField as the encoder while keeping it frozen. The suboptimal performance under this setting shows that training only the decoder is insufficient for scaling to our dataset, highlighting the importance of the encoder in building a promptable segmentation model. We then fine-tune the entire PartField encoder and observe notable improvements; however, its performance remains below that of our dual-branch encoder. This gap mainly stems from catastrophic forgetting—direct fine-tuning erodes the strong priors inherited from SAM, leading to degraded segmentation quality. In contrast, our dual-branch design preserves these 2D SAM priors in a frozen branch, while the learnable 3D-native branch acquires fine-grained 3D semantics from large-scale training. This architecture avoids forgetting effects and supports integrating auxiliary inputs (e.g., normals or RGB), resulting in the best overall performance.

### A.2.2 ABLATION OF TRAINING DATA

We next examine the effect of training data. Training only on PartNet (Mo et al., 2019) yields poor generalization, as the closed-world supervision fails to transfer to open-world scenarios. Using the curated data from Stage 1 (Sec. 3.5) improves generalization despite its limited diversity. Finally, augmenting with annotations from our model-in-the-loop pipeline provides further gains, confirming that large-scale and diverse supervision is essential for developing a scalable part segmentation model.

Figure 10: Visual comparison of native 3D part labels with 2D lifting part labels.

**Overall, the two ablations reveal complementary insights: the promptable model design ensures scalability, while diverse training data drives generalization across open-world settings.**

### A.2.3 ADVANTAGE OF 3D NATIVE DATA AND SCALING ANALYSIS

In Figure 10, we compare our native 3D part labels with the 2D-lifting results produced via community detection in Tang et al. (2024). The lifted 2D masks are frequently incomplete and overly fragmented due to heavy occlusions, making them unsuitable as reliable supervision in our framework. In contrast, our native 3D annotations maintain strong spatial coherence and accurately capture interior structures. Notably, even when the outer clothing of a character model is removed (second column), our 3D labels

still reveal clean and consistent part segmentation beneath the surface—an outcome fundamentally unattainable for 2D-lifting methods that only observe the visible exterior. The advantage of precise interior part segmentation is also illustrated in Figure 9.

To further evaluate the effect of large-scale native 3D training, in Figure 11, we present the scaling curve of PartSAM, showing how segmentation performance improves with increasing training data size, from 40k to 500k shapes. The results indicate a consistent improvement in both interactive and automatic segmentation, suggesting that PartSAM continues to benefit from large-scale 3D-native supervision. Notably, the performance gains persist even at the largest data size, demonstrating that the model is scalable and benefits from large-scale 3D-native supervision.

### A.2.4 APPLICATION

Benefiting from the powerful zero-shot ability, PartSAM can be applied in various downstream tasks and significantly outperforms prior works. Here, we showcase two representative tasks.

**Interior Part Editing.** PartSAM demonstrates strong performance in segmenting interior 3D structures, enabling detailed manipulation of segmented parts. Once a part is accurately segmented, it can undergo various types of edits, including material modifications (e.g., texture or color adjustments) and spatial changes, such as repositioning or reorienting within the 3D space. These capabilities make PartSAM particularly useful for applications requiring fine-grained control over complex 3D models, such as design and manufacturing processes.

**Amodal Part Segmentation** AI-generated meshes are often presented as a unified whole, with initial segmentation typically resulting in fragmented parts. Due to Part-

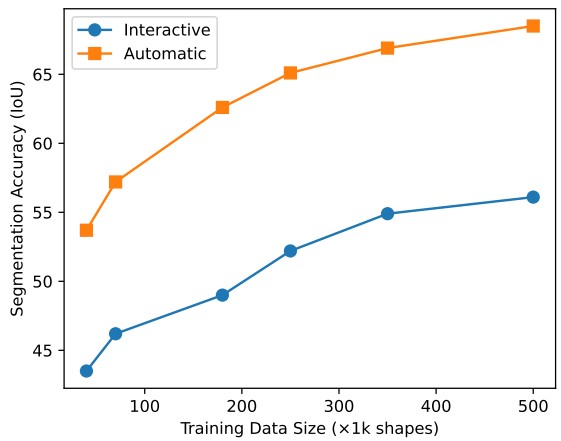

Figure 11: Scaling curve of PartSAM with respect to training data size. The plot shows segmentation accuracy (IoU for interactive segmentation with 1 prompt point and automatic segmentation) as the training data increases from 40k to 500k shapes.

SAM's strong generalization capabilities, it can accurately segment complex structures even when they are incomplete or fragmented. By integrating PartSAM with the part completion model HoloPart (Yang et al., 2025), these segmented fragments can be seamlessly reconstructed, achieving a comprehensive and complete decomposition of the mesh. This combined approach ensures that the segmented parts are not only accurate but also fully realized, overcoming the limitations of standard segmentation techniques.

### A.2.5 RESULTS ON PARTOBJAVERSE-TINY WITH MESH CONNECTIVITY

In the evaluation of automatic segmentation on PartObjaverse-Tiny (Section 4.2), we compare PartField (Liu et al., 2025) and our method in a setting where mesh connectivity is disabled. The rationale behind this choice is further explained through the quantitative results in Table 4. Rather than training PartField, we use a randomly initialized model to generate the feature field and apply agglomerative clustering to produce segmentation results with mesh connectivity (PartField (B) in Table 4). Interestingly, the segmentation performance is significantly better than that of the pre-trained PartField when mesh connectivity is disabled (PartField (A)), with a 27% improvement.

This result suggests that the mesh connectivity in PartObjaverse-Tiny (Yang et al., 2024b) contains rich prior knowledge about part labels, owing to the artistically crafted nature of the dataset. Consequently, using mesh connectivity for clustering in this dataset leaks ground-truth label information, leading to an unfair evaluation. Moreover, for shapes that require segmentation (such as AI-generated meshes), well-defined mesh connectivity is often absent. Therefore, we disable mesh connectivity in our main experiment to ensure a more representative evaluation.

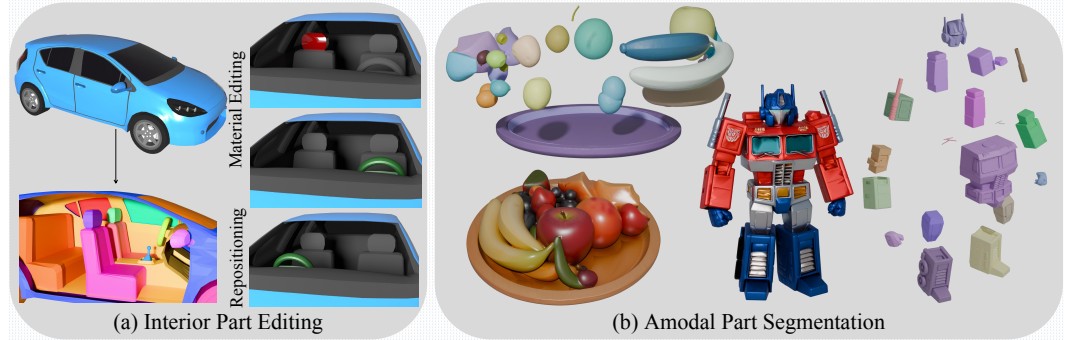

Figure 12: Application of PartSAM (a) Empowered by native 3D training data, PartSAM accurately segments interior parts and enables material editing and repositioning. (b) Combined with the recent part completion model HoloPart (Yang et al., 2025), PartSAM supports amodal part segmentation, achieving complete decomposition of AI-generated meshes.

Moreover, as indicated by the results in the last row of Table 4, we find that PartSAM can also benefit from mesh connectivity information. After getting the per-face segmentation labels using our segment-every-part mode, we apply graph cuts to refine the segmentation, following Boykov et al. (2001). Specifically, we construct a mesh graph where each face is treated as a node, and edges represent adjacency relationships between faces. The graph cut algorithm minimizes an energy function that balances two key components: (1) a data term which measures the disagreement between the refined label

Table 4: Quantitative comparison of automatic segmentation on PartObjaverse-Tiny (Yang et al., 2024b) under different settings. We report the mean IoU on instance-level labels.

| Method | connectivity | Training | IoU |
|---|---|---|---|
| PartField (A) | | ✓ | 51.5 |
| PartField (B) | ✓ | | 65.6 |
| PartField (C) | ✓ | ✓ | 79.2 |
| Ours (A) | | ✓ | 69.5 |
| Ours (B) | ✓ | ✓ | **81.3** |

assignment and the initial per-face predictions, and (2) a smoothness term that penalizes large label changes between adjacent faces, encouraging spatially coherent boundaries. By minimizing this energy function, this post-processing ensures adjacent faces have consistent labels, thereby improving segmentation accuracy. By using connectivity information in this way, PartSAM outperforms PartField under this setting (Ours (B)), further demonstrating the flexibility of our method.

### A.2.6   COMPLEXITY ANALYSIS

We present the complexity analysis in Table 5, reporting both the inference time on a single NVIDIA H20 GPU and the number of network parameters. Compared with lifting-based methods that require per-shape optimization (Tang et al., 2024; Yang et al., 2024b), PartSAM achieves substantially faster inference due to its feed-forward design. Compared with PartField (Liu et al., 2025), PartSAM incurs only a slight increase in inference time for segmenting an entire shape. Although our encoder is built upon PartField, it eliminates the need for per-face dense feature sampling and clustering. Furthermore, our decoder is lightweight, ensuring that the overall inference time remains well-controlled. In addition, we further report the runtime and performance under different numbers of prompt points (1024/512/256). The results show that PartSAM delivers consistently strong performance across all settings, indicating that our method is robust to the choice of prompt density. These results demonstrate that our method effectively balances computational cost and performance.

### A.2.7   CROSS-SHAPE CONSISTENCY.

We follow the evaluation protocol of Liu et al. (2025) and use functional maps (Ovsjanikov et al., 2012) to compute correspondences between pairs of shapes in Figure 13. For similar quadruped shapes (first row), both PartField and our PartSAM achieve reasonable cross-shape consistency, yet ours yields clearer correspondences in fine-grained regions such as the tail. When the structural gap becomes larger (second row), our method remains robust—producing correct correspondences on challenging parts like the hands—whereas PartField fails and yields entirely incorrect mappings.

These results highlight the stronger part-aware consistency of our learned features under large-scale native 3D supervision.

### A.2.8 FAILURE CASES AND LIMITATIONS.

We present failure cases of PartSAM in Figure 14, and provide additional results on diverse meshes in Figures 17 and 18, where failure cases are highlighted using red dashed boxes. Although PartSAM benefits from large-scale native 3D supervision, the diversity of existing 3D datasets (Deitke et al., 2023b;a) remains limited compared with high-coverage 2D datasets such as SA-1B (Kirillov et al., 2023). Consequently, structures that rarely appear during curation may not be accurately segmented at test time. For example, in the first row of Figure 14, PartSAM fails to extract the engraved letters carved into surfaces—these surface-level markings never appear as valid parts in current 3D pipelines. SAMesh can sometimes detect such tiny engraved regions due to its direct use of SAM outputs, though its multi-view masks may be inconsistent and introduce noise. PartField, which relies on feature space clustering, is even less able to recover such fine-grained details. Another type of failure arises when an object lacks a clear semantic structure, as exemplified by the coral sculpture in the second row. In these cases, both PartSAM and existing baselines struggle to produce meaningful decompositions because the object itself does not possess a coherent part hierarchy.

These cases highlight a limitation of current 3D data curation: the class-agnostic nature of part annotation, combined with the inherent ambiguity of open-world part definitions, leads to a long-tail distribution where many part types appear only rarely. Such an imbalance makes it difficult for the model to distinguish the rare, minority parts. This issue becomes even more pronounced on AI-generated meshes or coarse real-world scans, where geometric irregularities, incomplete structures, or hallucinated details further amplify the ambiguity of part boundaries and exacerbate the difficulty of consistent part decomposition.

Meanwhile, although PartSAM achieves state-of-the-art performance in class-agnostic part seg-

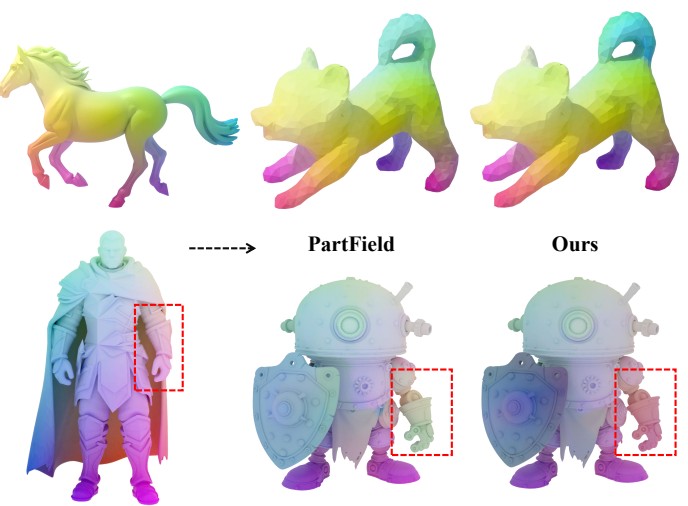

Figure 13: Point-to-point correspondences obtained by Functional Maps (Ovsjanikov et al., 2012) using features of PartField (Liu et al., 2025) and our encoder as input.

mentation, similar to 2D SAM (Kirillov et al., 2023; Ravi et al.), it cannot directly produce semantic labels for these masks, which are important for some downstream tasks. One possible future direction to address this limitation is to generate large-scale training datasets consisting of 3D shapes, part annotations, and corresponding semantic labels. This could potentially be achieved by utilizing PartSAM's interactive segmentation capabilities, where users provide feedback on segmented parts, facilitating the assignment of semantic categories.

### A.3 EVALUATION ON 3DCOMPAT++

To comprehensively evaluate PartSAM, we compare it with PartField (Liu et al., 2025) and our ablation variant trained without the second-stage native 3D data on a recent dataset, 3DCoMPaT++ (Slim et al., 2025). Since the part annotations in 3DCoMPaT++ are semantic-level labels, both the baselines and PartSAM experience performance drops compared with the results on instance-level datasets. Nevertheless, PartSAM still significantly outperforms PartField and also surpasses our ablation variant trained without the second-stage native 3D supervision, highlighting the importance of large-scale 3D data in achieving strong generalization. This performance gap aligns with the characteristics of the dataset. 3DCoMPaT++ contains extremely fine-grained part annotations, which pose challenges for

Table 5: Complexity analysis. We compare the time of automatic segmentation, the time of interactive segmentation, and the number of trainable network parameters.

| Methods | Time | Params | Performance |
|---|---|---|---|
| SAMesh | $\sim$ 7min | / | 56.9 |
| SAMPart3D | $\sim$ 15min | 114M | 53.5 |
| PartField | $\sim$ 10s | 106M | 51.5 |
| Ours | $\sim$ 12s (Encoder: 1.2s, Decoder: 0.01s$\times$1024 points) | 118M | 69.5 |
| | $\sim$ 9s (768 points) | | 68.8 |
| | $\sim$ 7s (512 points) | | 67.9 |
| | $\sim$ 4s (256 points) | | 65.2 |

clustering-based methods like PartField—whose clustering step struggles to reliably separate small or rare parts. In contrast, our promptable decoder naturally produces masks across multiple granularities, enabling the model to recognize and segment diverse structures within a unified framework. This allows PartSAM to handle the fine-grained decomposition required in 3DCoMPaT++.

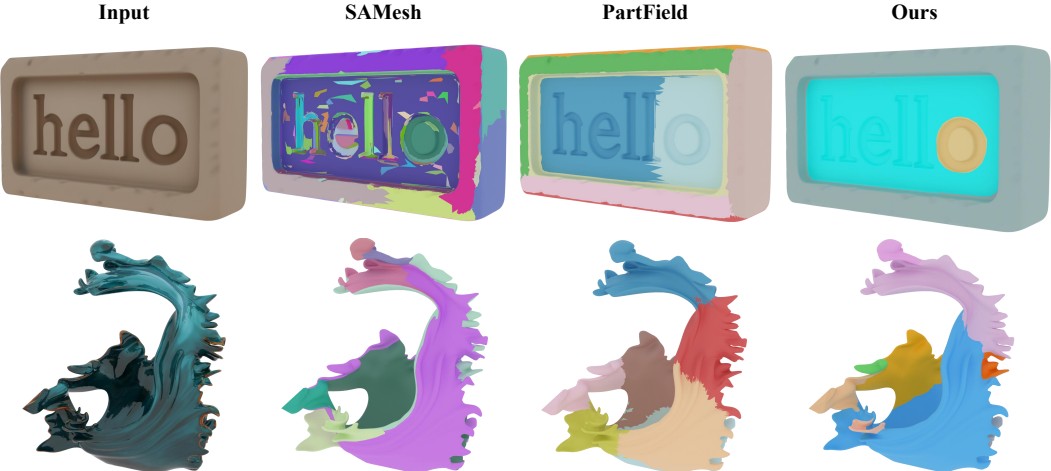

Figure 14: Failure cases of PartSAM.

## A.4 ADDITIONAL VISUALIZATION

**Visualization of Annotated Part Labels** We visualize the part labels annotated using the Model-in-the-loop pipeline in Figure 15 and observe that our pipeline consistently produces high-quality part labels through the proposed filtering rule. For instance, for the gun in the first row, the first prompt results in an IoU that exceeds the threshold, making it a valid annotation. The boat part in the first row has a low IoU after the first iteration, but by the 10th iteration, the IoU surpasses the threshold, making it a valid annotation. In contrast, for the parts in the last row, the IoU never exceeds the threshold in any iteration, so no annotation is generated.

**Visualization of Segmentation Results** Additional segmentation results of PartSAM are shown in Figure 16, where it consistently delivers impressive performance across different kinds of shapes. To provide a more comprehensive picture of PartSAM's behavior, we also present randomly sampled results in Figure 17 and Figure 18, where failure cases are highlighted with red dashed boxes.

Table 6: Quantitative comparison of automatic segmentation on 3DCoMPAT++ (Slim et al., 2025). The **best** scores are emphasized in bold. We report the mean IoU.

| Category | Method | Coarse | Fine-grained | Avg |
|---|---|---|---|---|
| Plane | PartField | 47.3 | 30.5 | 38.9 |
| | Ours w/o Model-in-the-loop Annotation | 57.7 | 42.6 | 50.2 |
| | Ours | **59.4** | **47.5** | **53.5** |
| Car | PartField | 40.5 | 20.7 | 30.6 |
| | Ours w/o Model-in-the-loop Annotation | 54.1 | 33.5 | 43.8 |
| | Ours | **57.9** | **37.7** | **47.8** |
| Bag | PartField | 51.9 | 41.4 | 46.7 |
| | Ours w/o Model-in-the-loop Annotation | 61.2 | 52.7 | 57.0 |
| | Ours | **66.9** | **55.3** | **61.1** |
| Coat Rack | PartField | 48.0 | 44.7 | 46.4 |
| | Ours w/o Model-in-the-loop Annotation | 58.8 | 46.3 | 52.6 |
| | Ours | **62.5** | **50.2** | **56.4** |
| Toilet | PartField | 44.2 | 38.7 | 41.5 |
| | Ours w/o Model-in-the-loop Annotation | 58.4 | 62.2 | 60.3 |
| | Ours | **60.3** | **65.7** | **63.0** |

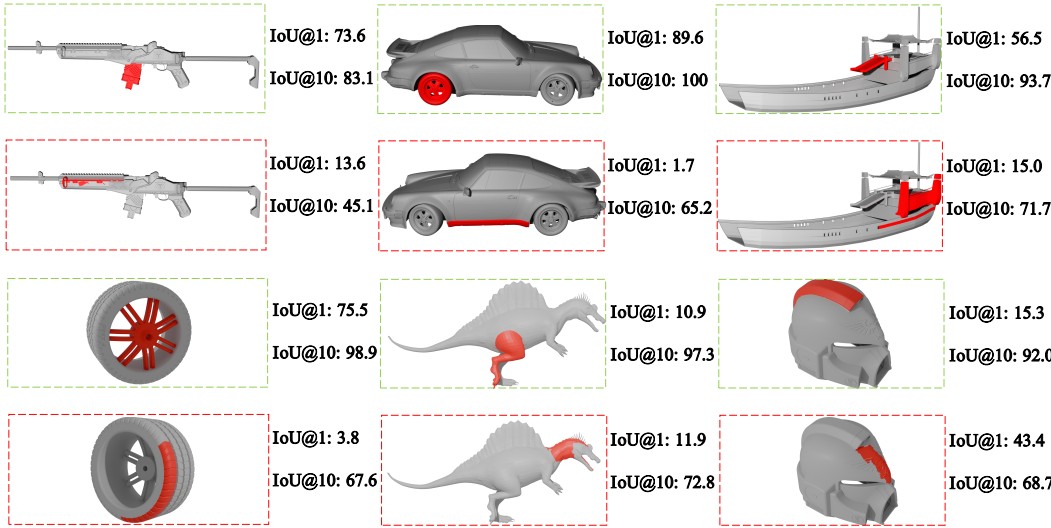

Figure 15: Example of part labels generated by PartField (Liu et al., 2025). Masks in green dashed boxes satisfy our IoU-based criteria and are retained as training data, while masks in red dashed boxes are filtered out by our model-in-the-loop strategy.

**PartObjaverse-Tiny**   **AI-generated**

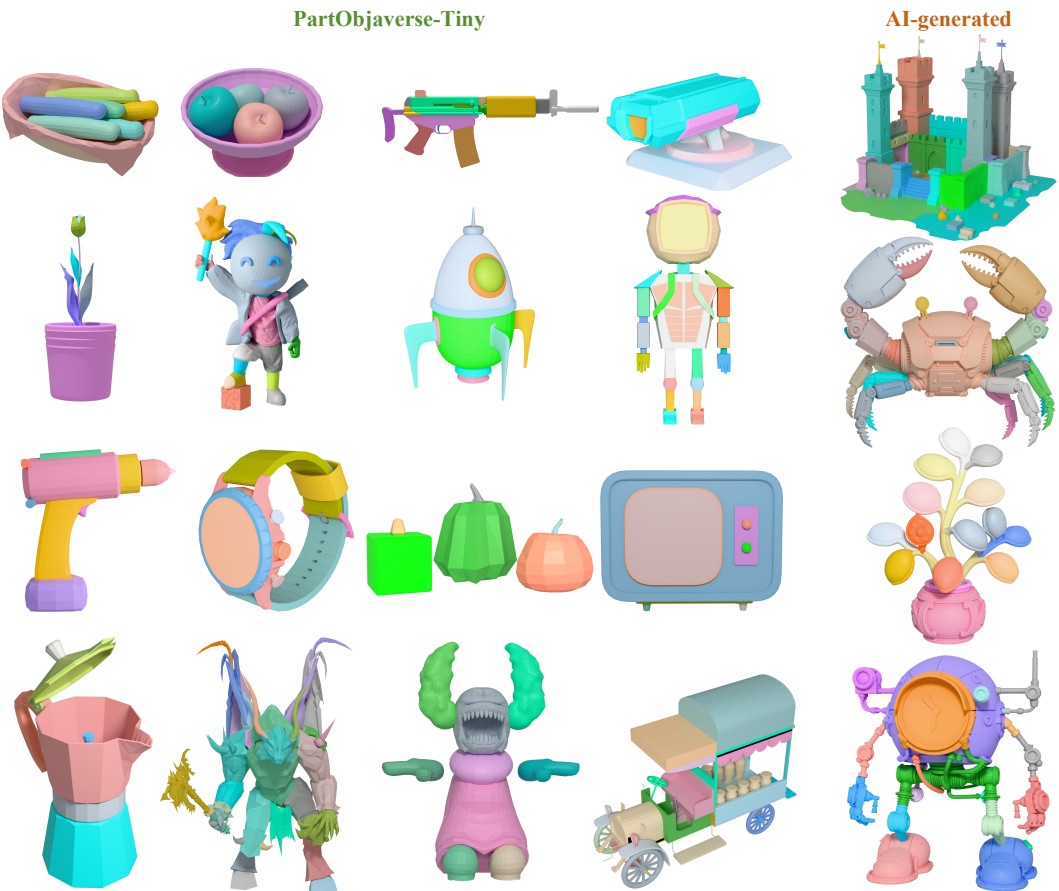

Figure 16: Additional visualization of PartSAM's automatic segmentation results on PartObjaverse-Tiny (Yang et al., 2024b) and AI-generated shapes (Lai et al., 2025).

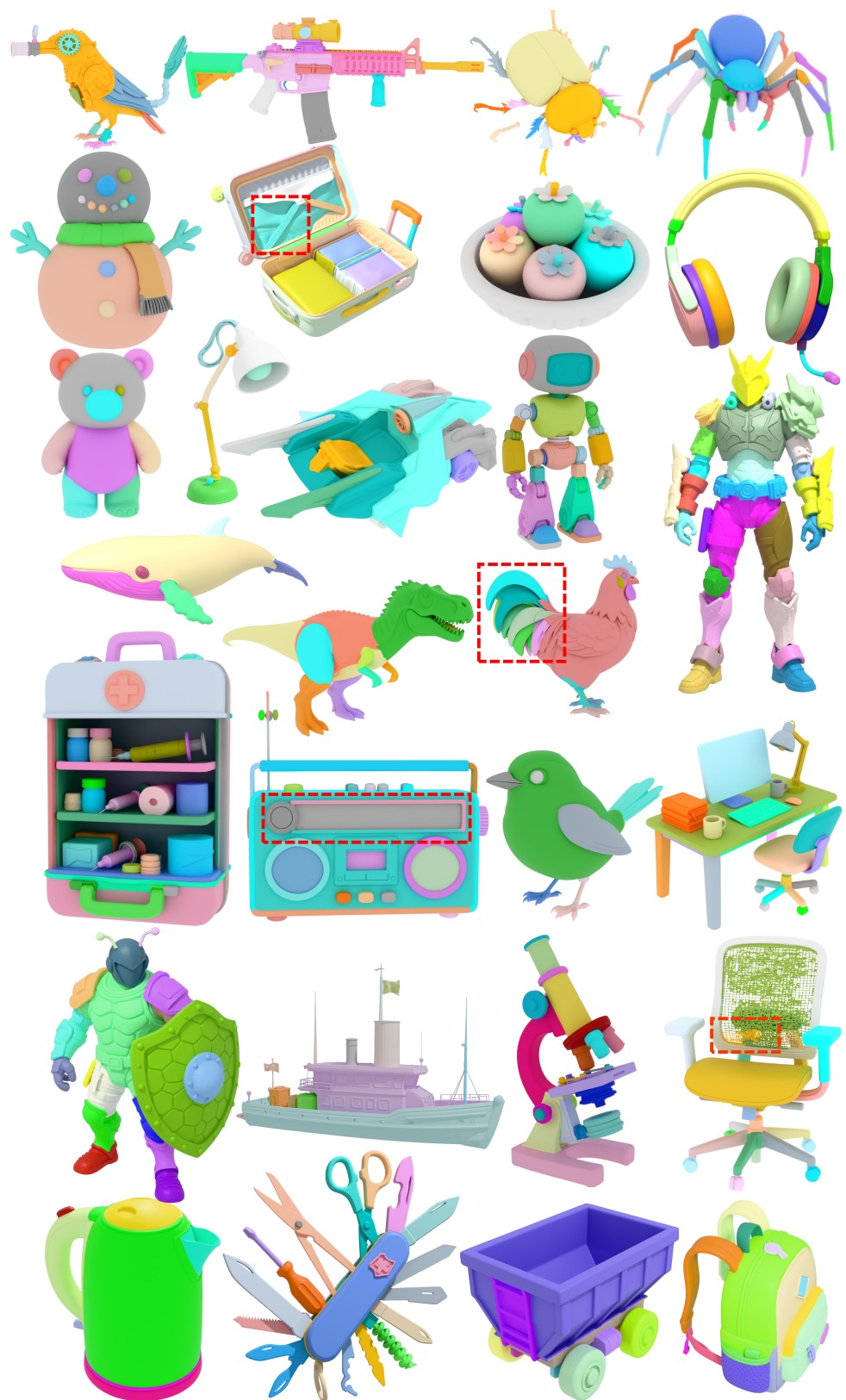

Figure 17: Randomly sampled automatic segmentation results on AI-generated shapes (Lai et al., 2025). Failure cases are highlighted with red dashed boxes.

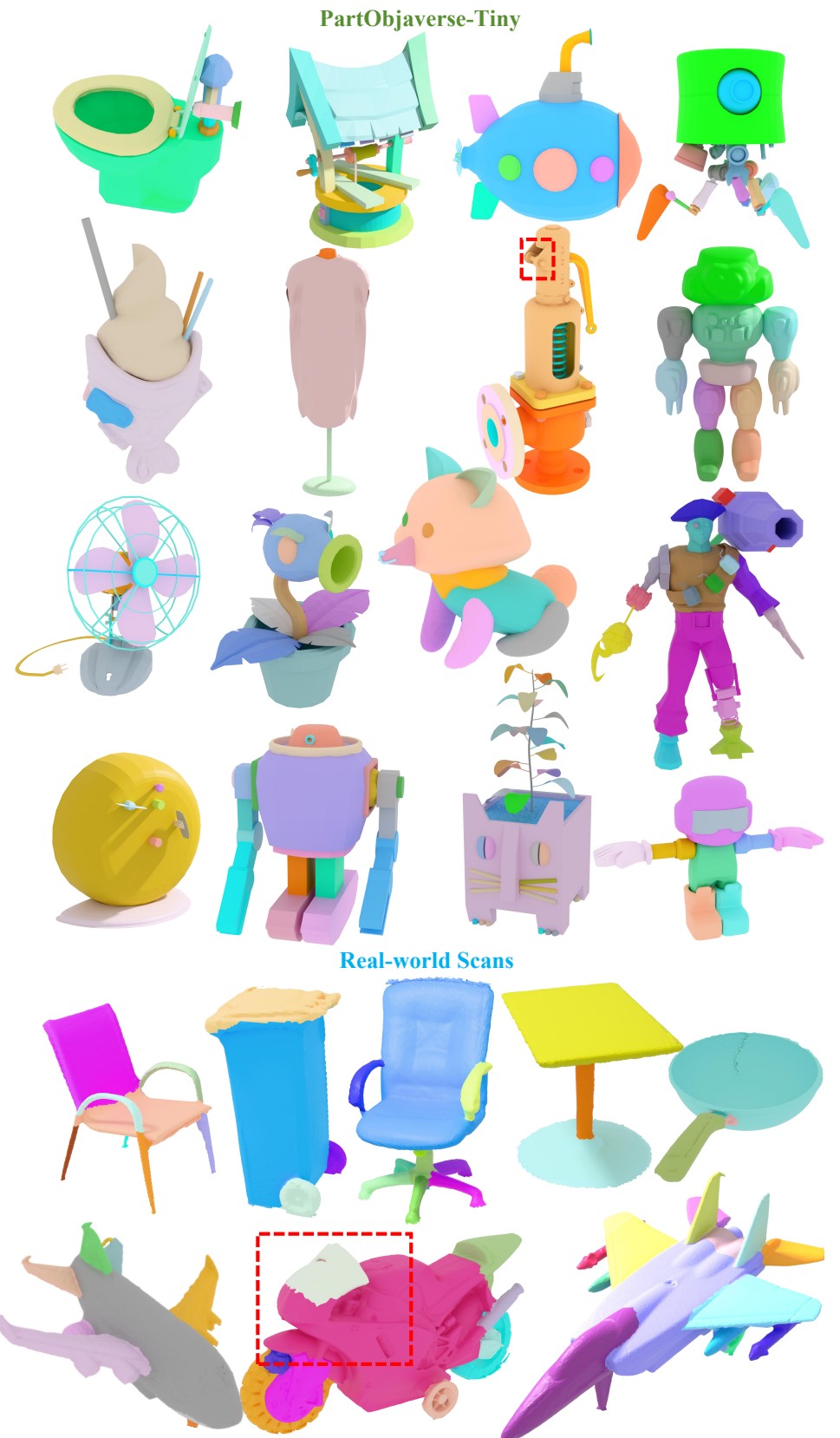

Figure 18: Randomly sampled automatic segmentation results on Yang et al. (2024b) and real-world scans (Choi et al., 2016; Wu et al., 2023). Failure cases are highlighted with red dashed boxes.

