# OpenReview forum: "PartSAM: A Scalable Promptable Part Segmentation Model Trained on Native 3D Data"
_ICLR.cc/2026/Conference — ICLR 2026 Poster_

### Official Review · Reviewer_faSk · 2025-10-29

**Soundness:** 3
**Presentation:** 3
**Contribution:** 3
**Rating:** 6
**Confidence:** 4

**Summary:**

The authors propose PartSAM, a model for 3D part segmentation. Specifically, the PartSAM contains two essential components: encoder with initialized weight from previous work that inherits 2D foundation prior, and further train with native 3D data; and scaling the 3D data with model-in-the-loop annotation. Then the scaling of data results in a very powerful model for 3D part segmentation, with great in-the-wild generalization ability, and better performance comparing to previous methods.

**Strengths:**

The paper has very clear motivation and intuition for each method components. For example, using PointSAM backbone as initialization to adopt "2D SAM image prior" is a clever design and naturally fits.

The authors also conducted extensive experiments to compare with baselines, and also ablations to show the validness of each proposed components.

The overall presentation of the paper is good and well-structured, with clear transition from motivation, to method design, to experiment part to show the validness of the proposed method.

**Weaknesses:**

I think the main weakness is that some details of the method / the system is not that clear in the current paper. I'd like authors to add more explanations on these parts:

1. What's the detailed model architecture to generate mask given point prompt, in the architecture of the network and also the shape format of all the input-output here.

2. What's the mechanism of incorporating previous point prompt? Both as a positive point prompt or a negative point prompt.

3. What's the detailed process of the IoU head in the network?

In general I think the network architecture and process can be elaborated more, to make it more clear to the readers.

**Questions:**

Please see the weaknesses above. Besides that, some other questions:

1. Will the non-clean meshes, or meshes with broken geometry in the dataset harm the training process? For example many assets in Objaverse are "not semantically meaningful" assets with e.g. a flat thin surface. Is there any process to avoid using these data, or these data are causing some negative effect.

2. It'd be good to show more qualitative results on "inner structure" of the model's output comparing to baselines, to showcase the advantage of trained with native 3D data.

3. It'd also be good to show some visualization results about the failure case.

In general I think this is a good paper, with aforementioned questions and weaknesses being answered/addressed, I'll raise my score accordingly.

---

> ### Author Response · Authors · 2025-11-23
>
> We thank the reviewer for their time in reviewing our paper and for their insightful comments. We are glad that the reviewer appreciated *the clear motivation behind our design choices, the extensive experiments, and the well-structured presentation*. Below, we tried our best to address the remaining concerns raised by the reviewer:
> > The detailed model architecture and IoU head
>
> We add a description of the complete details of our segmentation pipeline in Appendix A.1.1. To ensure clarity, we also include an explicit reference to this section in Section 3 so that readers can easily locate the full pipeline description.
>
> > Mechanism of incorporating previous point prompt
>
> We describe the mechanism in detail in Appendix A.1.2: In the first round of interaction, we input the first prompt point and obtain the initial mask result. In subsequent rounds, we concat the last predicted mask logits with coordinates and adopt the same sampling-and-grouping strategy as in Equation (1) to extract dense mask embeddings $F_p \subseteq \mathbb{R}^{N_c \times C}$. These dense embeddings are directly added to the input embeddings $F_c$, allowing the model to incorporate information from previous prompts. For the new prompt point, we distinguish positive and negative prompts by adding different learnable prompt-type embeddings to $F_p$.
>
> > Non-clean Training Data
>
> Our dataset is constructed through a fully automated two-stage curation pipeline. We first filter out low-quality assets (e.g., scanned objects, abnormal part sizes, and shapes with too few or too many parts). During this step, meshes with broken geometry, non-manifold surfaces, and assets that are not semantically meaningful have been filtered. We then refine the remaining data using connectivity-aware clustering and a model-in-the-loop filtering process at both part and shape levels. Only shape–part pairs that pass all criteria are retained. This ensures high-quality 3D-native supervision at scale. Full details of the curation steps are provided in Appendix A.1.3.
> Our pipeline is fully automated to ensure scalability, so a small number of imperfect cases may remain without manual intervention, though their proportion is negligible. Importantly, the scaling curves in Figure 11 show consistent performance gains as data size increases, directly indicating that the overall data quality is sufficient and continues to benefit the model.
>
> > More qualitative results on inner structure
>
> We have presented more qualitative results on inner structure and occluded parts in Figure 9, where PartSAM consistently outperforms 2D-lifting-based methods.
>
> > Failure cases
>
> We present failure cases of PartSAM in Figure 14, and additional results on diverse meshes in Figures 17 and 18, where failure cases are highlighted using red dashed boxes. These cases mainly arise from structures that are rare in current 3D datasets (e.g., recessed surface details and coarse structures in AI-generated meshes), which PartSAM and other 3D methods are not trained on. We also observe failures on shapes that lack a clear semantic decomposition. Detailed analysis can be found in Appendix A.2.8.

---

### Official Review · Reviewer_9Mx7 · 2025-10-31

**Soundness:** 2
**Presentation:** 2
**Contribution:** 2
**Rating:** 4
**Confidence:** 4

**Summary:**

This paper introduces PartSAM, a promptable 3D part segmentation model trained on native 3D data that outperforms state-of-the-art methods on both interactive and automatic part segmentation tasks. The work also contributes a large-scale 3D part dataset comprising 180K shapes created by merging existing part labels into semantically consistent parts, along with 320K shapes annotated using pseudo-labels.

**Strengths:**

1. PartSAM achieves significant quantitative improvements over existing methods on its benchmarks for both interactive and automatic part segmentation tasks.
2. In the data curation process, the merging criteria that exclude assets failing to form semantically meaningful parts is an excellent filtering strategy. Although automatic part segmentation is performed in a class-agnostic manner, ensuring that segmented parts can be assigned appropriate semantics remains valuable and well-motivated.

**Weaknesses:**

1. PartObjaverse-Tiny contains only 200 shapes across 8 categories, which is insufficient to convincingly demonstrate model performance. While evaluation on the PartNetE dataset (1,906 shapes) provides additional evidence, evaluating on larger and more diverse benchmarks such as 3DCOMPAT++ [1] would significantly strengthen the claims.
2. Although the ablation study in the Appendix demonstrates performance improvements brought by the model-in-the-loop annotation, the results are only evaluated on the small PartObjaverse-Tiny dataset. The paper would benefit from more comprehensive analysis demonstrating annotation quality. Given that the data curation process emphasizes semantically meaningful parts, I believe 3DCOMPAT++'s part definitions would be more appropriate for evaluation, as they possess stronger semantic grounding.

[1] 3DCOMPAT++: A comprehensive dataset for 3D object understanding with fine-grained part annotations

**Questions:**

1. Given the weakness 1 and 2, would it be possible to evaluate performance on the car and airplane categories (68 meshes in total) from 3DCOMPAT++ [1] and compare the results with PartField at this stage? It would be particularly valuable if the evaluation could be performed on both fine-grained and coarse part levels.
2. PartField demonstrates emergent cross-shape consistency when exploring feature space. Could you provide both qualitative and quantitative (if possible) results showing whether PartSAM exhibits similar cross-shape consistency, along with comparisons to PartField?
3. As mentioned in the model-in-the-loop annotation section, PartField suffers from limited controllability. In my experience, it sometimes cannot even segment all four wheels of a bus. Given this limitation, is using PartField outputs as candidate labels truly a reliable choice to obtain semantically meaningful annotations? Are there specific technical safeguards in place to prevent such failures from compromising pseudo-label quality?
4. Previous 2D-lifting approaches often suffer from limitations such as 3D inconsistency in 2D models. The motivation for a native 3D solution should therefore be to avoid such error modes inherent in 2D models. Given this context, I am curious why it is necessary to retain 2D priors from SAM in the dual-branch encoder design?

[1] 3DCOMPAT++: A comprehensive dataset for 3D object understanding with fine-grained part annotations

---

> ### Author Response · Authors · 2025-11-23
>
> We thank the reviewer for their time in reviewing our paper and for their insightful comments. We are grateful that the reviewer found the *strong performance improvements, the excellent filtering strategy, and the well justified motivation*. Below, we tried our best to address the remaining concerns raised by the reviewer:
>
> > Evaluation on 3DCOMPAT++
>
> As suggested, we evaluated automatic segmentation on the two 3DCoMPaT++ categories (68 meshes). Because 3DCoMPaT++ uses fine-grained semantic labels, all methods show lower accuracy than on instance-level datasets. Even so, PartSAM clearly outperforms PartField and our ablation without second-stage training, illustrating the benefit of large-scale 3D-native training. The results are consistent with the dataset’s characteristics: its extremely fine-grained parts are difficult for clustering-based approaches, while PartSAM’s decoder naturally produces multi-granularity masks that better capture such structures.
>
> | Category | Method                               | Coarse | Fine-grained | Avg  |
> |----------|----------------------------------------|--------|--------------|-------|
> | Plane    | PartField                              | 47.3   | 30.5         | 38.9  |
> |          | Ours w/o Model-in-the-loop Annotation  | 57.7   | 42.6         | 50.2  |
> |          | **Ours**                               | **59.4** | **47.5**     | **53.5** |
> | Car      | PartField                              | 40.5   | 20.7         | 30.6  |
> |          | Ours w/o Model-in-the-loop Annotation  | 54.1   | 33.5         | 43.8  |
> |          | **Ours**                               | **57.9** | **37.7**     | **47.8** |
>
> > Cross-shape consistency
>
> We follow PartField’s protocol and use functional maps to compute cross-shape correspondences (Figure 13). For similar quadruped shapes, both methods show reasonable consistency, though PartSAM provides clearer matches in fine-grained areas such as the tail. When the structural difference increases, PartSAM remains robust and produces correct correspondences on challenging parts, while PartField fails entirely. These results demonstrate the stronger part-aware consistency of our features under large-scale 3D-native supervision.
>
> >  Is using PartField outputs as candidate labels truly a reliable choice to obtain semantically meaningful annotations?
>
> We agree that the raw PartField outputs are not reliable enough to be used directly. In the second stage of our curation pipeline, we therefore apply several filters to improve quality: First, we incorporate artist-defined connectivity information into PartField’s clustering process, enabling it to generate clean segments with clear boundaries. Second, our interactive annotation procedure introduces two levels of safeguards: a part-level filter that only accepts segments satisfying the interactive IoU standard, and a shape-level filter that retains only shapes containing more than five valid parts. After these filtering steps, we keep only the shape–part pairs that pass all criteria. Consequently, clustered masks without clear semantic meaning (e.g., wheels merged into the car body) are consistently removed during this process. Full details of the curation steps are provided in Appendix A.1.3.
> Our pipeline is fully automated to ensure scalability, so a small number of imperfect cases may remain, though their proportion is negligible. Importantly, the scaling curves in Figure 11 show consistent performance gains as data size increases, directly indicating that the overall data quality is sufficient and continues to benefit the model.
>
>
> > Necessities of 2D prior branch
>
> Thanks for the thoughtful question. We conduct an ablation study by training PartSAM entirely from scratch. The clear performance drop confirms that **the distilled 2D SAM features provide essential priors that significantly benefit 3D part segmentation**.
> | Method | IoU@1 | IoU@3 | IoU@5 | IoU@7 | Automatic |
> |----------|----------|----------|----------|----------|----------|
> | w/o pre-trained weights  | 48.3  | 73.6  | 79.1  | 80.5  | 60.5  |
> | Ours  | **56.1**  | **79.0**  | **84.7**  | **86.9**  | **68.5**  |
>
> These results can be explained by the structure of SAM, which consists of a strong encoder and a prompt-based decoder. In 2D-lifting approaches, the 3D inconsistency comes from the decoding stage, where 2D masks must be fused across views. PartSAM avoids this issue entirely—we do not use any 2D masks. Instead, we use only the encoder-level features already distilled into a 3D-aligned triplane field, which introduces no view-lifting artifacts. This design retains SAM’s useful representation priors while sidestepping the failure modes of 2D lifting.

---

> > ### Comment · Reviewer_9Mx7 · 2025-11-25
> >
> > Thank you for the detailed response and additional results. I appreciate the efforts to pursue semantically meaningful parts in this work, so I want to ensure that the data curation and evaluation are conducted properly for this purpose. The new results on 3D-CoMPaT++ are more convincing, and I encourage the authors to extend the evaluation to more categories in the future revision, as this dataset is better for assessing semantic quality. My concerns have been addressed, and I am raising my score to 6.

---

> > > ### Author Response · Authors · 2025-11-26
> > >
> > > We truly appreciate your reconsideration and are glad that our rebuttal addressed your concerns. Your suggestion to extend the evaluation to more categories is highly valuable, and we plan to incorporate this direction in future revisions. Thank you again for your constructive feedback.

---

### Official Review · Reviewer_zpAD · 2025-11-02

**Soundness:** 3
**Presentation:** 3
**Contribution:** 3
**Rating:** 6
**Confidence:** 3

**Summary:**

This paper introduces PartSAM, a 3D part segmentation model that avoids lifting 2D masks, supports interactive prompts, and offers a “segment-every-part” mode for full-shape decomposition. Architecturally, it combines a triplane dual-branch encoder (one frozen to preserve SAM-derived 2D priors, one learnable for native 3D) with a prompt-guided decoder, enabling controllable, geometry-aware segmentation including interior structures. A model-in-the-loop pipeline scales supervision to >5M shape–part pairs from large 3D asset repositories, yielding diverse, fine-grained labels. Experiments show decisive SOTA gains over Point-SAM/PartField on PartObjaverse-Tiny and PartNet-E.

**Strengths:**

I like the paper's architecture–data efforts (SAM-like decoder + dual-branch triplane encoder + large 3D supervision pipeline) yields generalizable segmentation that extends to interior parts. Although no novel technical contributions, but I do like the whole pipeline and believe such model-data efforts are good to community if open-sourcing.

The evaluation is extensive, including interactive/automatic settings, ablations, complexity analysis. The results show consistent over strong baselines, including SAMPart3D and PartField.

**Weaknesses:**

The part-level are ambitious or the part definition is unclear. For example, the left-second human-horse case in Figure 1 segment the tail into two weird chunks, while the hair and the horse head are not separate. Also, the current multi-scale is generated by varying NMS
threshold value, and also the beginning is by using PartField along with different clustering number. I feel this is a fundamental limitations for the current setup.

The performance of SAMesh is actually quite good as well, where the proposed method is superior compared to SAMPart3D and PartField, as the boundaries is more clear. However, SAMesh also has clear boundaries. The proposed method is more fine-grained in large pieces, while SAMesh tend to segment into large pieces, where I feel can tune the 2D segmentation model fine-grained level to adjust this point. Also, the grenade in SAMesh is very fine-grained. Therefore, I do not believe the proposed method is absolutely better than SAMesh. For this point, I actually want to see more qualitative comparisons between the proposed method and SAMesh with randomly sampling. Not cherry-pick, but randomly sampling can give reader better understanding of the performance differences. Also, there is no need for the proposed method absolutely better than SAMesh as two methods are driven from very different technical backbones. I just encourage the author to conduct the comparisons more systematics, and show the limitations for both the SAMesh and the proposed method. This will not harm the paper at all.

**Questions:**

Plz address the concerns raised in the weakness and ethics concerns sections.

**Details Of Ethics Concerns:**

This paper introduces five million native 3D shape–part pairs built upon Objaverse(-XL). However, many objects in Objaverse(-XL) are subject to commercial license restrictions, and some are no longer downloadable from their original sources, such as those previously hosted on GitHub. I wonder how the authors plan to release the data or model, and under what license. If this paper only showcases results from closed-source efforts, its contribution will be significantly limited.

---

> ### Author Response · Authors · 2025-11-23
>
> We thank the reviewer for their time in reviewing our paper and for their insightful comments. We are glad that the reviewer appreciated our *architecture–data design, the generalization to interior parts, and the value of the full pipeline to the community*. We also thank the reviewer for recognizing *the breadth of our evaluation and the consistent improvements over baselines*. Below, we tried our best to address the remaining concerns raised by the reviewer:
>
> > Part definition
>
> Open-world part segmentation does not assume a fixed taxonomy, and many meshes—especially AI-generated or stylized ones—contain genuinely ambiguous structures. In the human–horse example, the long tail has two disconnected lobes, so the model produces two fine-grained subparts. This reflects ambiguity in the geometry rather than model instability; our goal is to provide consistent groupings across different granularities, not to enforce a single canonical answer. We have added this discussion into Appendix A.2.8. Meanwhile, PartSAM handles such cases better than prior methods for two reasons.
> (1) Training on native 3D data gives the model stronger spatial awareness, enabling coherent 3D-consistent decompositions that 2D-lifting methods struggle to achieve (see Fig. 7–9).
> (2) PartSAM naturally supports multi-scale and interactive refinement—its decoder outputs masks at multiple granularities, and users can further refine ambiguous regions via prompt points.
>
> > Multi-scale segmentation
>
>
> Thanks for the question. We clarify the difference in two points:
> **PartSAM does not use PartField's clustering during inference:** Clustering is used only in the data curation stage to generate pseudo-labels. After training, PartSAM operates purely as a native 3D feed-forward model and no longer relies on PartField or any clustering procedure.
> **Difference between our multi-scale segmentation and PartField's clustering:** PartSAM’s parallel decoder directly outputs a rich set of part-aware masks at multiple granularities, meaning that all parts are already well-formed before NMS; NMS merely selects a non-overlapping subset. In contrast, PartField must impose a fixed clustering number to produce parts. As shown in Figure 8, this clustering constraint often causes over-segmentation and coarse boundaries, whereas PartSAM yields reasonable and coherent results across scales. Moreover, since PartSAM naturally produces multi-scale masks, hierarchical segmentation can be obtained through other simple controls, such as restricting mask sizes before applying NMS.
>
> > Comparison with SAMesh and discussions
>
> Thanks for the constructive comments. We agree that SAMesh can also produce sharp and visually clean boundaries, as it directly inherits the 2D SAM mask predictions. To provide a more systematic comparison, we have added additional qualitative results in Figure 9 (interior structure) and Figure 14 (failure cases).
> **SAMesh limitations for 3D part segmentation:** While SAMesh performs well on prominent surface features, this 2D-driven granularity can also lead to over-fragmentation of semantically coherent 3D parts, and multi-view rendering causes occluded or interior structures to be missed entirely. In contrast, PartSAM learns directly from native 3D supervision and produces more coherent and complete decompositions, including interior parts. A more detailed analysis is provided in Section 4.3.
>
> > Data release
>
> We appreciate the reviewer’s concern regarding licensing and dataset accessibility. Our work does not redistribute any raw Objaverse(-XL) assets, nor do we host or mirror any original 3D objects. All assets must be obtained through the official Objaverse(-XL) URLs under their original licenses. What we release are only the derived shape–part annotations (mesh vertice-level part masks), which contain no copyrighted geometry or textures. Such derivative annotations are permitted as long as they do not include or reconstruct the original assets. To ensure full compliance and reproducibility, we will release:
> (1) all part annotations under a permissive license (CC-BY-4.0),
> (2) data processing script used in the two-stage data curation pipeline so users can reproduce the dataset with their own data
> (3) all model weights and training code.
> With this release plan, our work is fully open, reproducible, and license-compliant.

---

> > ### Comment · Reviewer_zpAD · 2025-11-24
> > **Reviewer response**
> >
> > I appreciate the authors’ responses.
> > However, the added failure cases are not sufficiently extensive (A.2.8). Could the authors provide two pages of randomly sampled results generated by PartSAM? This would help readers better understand the actual capabilities of the proposed pipeline. I would be happy to potentially raise my score with this additional evidence.

---

> > > ### Author Response · Authors · 2025-11-26
> > >
> > > Thank you for the constructive follow-up suggestion. Following the reviewer’s request, we have added two full pages of randomly sampled automatic segmentation results generated by PartSAM, with failure cases highlighted using red dashed boxes:
> > >
> > > Figure 17: results on a diverse collection of AI-generated meshes;
> > >
> > > Figure 18: results on artist-created meshes and real-world scans.
> > >
> > > These examples cover a wide range of geometries and difficulty levels. We believe they provide a clearer and more comprehensive picture of PartSAM’s behavior, including both its strengths and its limitations. We hope these additional results address the reviewer’s concern and further clarify the actual capabilities of our pipeline.

---

> ### Comment · Reviewer_zpAD · 2025-11-28
> **Reviewer response**
>
> I appreciate the efforts made by the authors, thanks! If the results are indeed randomly sampled, plz mention it in the Figure 17 & 18 title -- as it is an important criteria.
>
> It seems I cannot adjust my score at this moment. I will make sure raise to 8 if the system later re-allow -- if the system cannot, I will make sure delivery this point to the AC. Thanks and congrats to this good work!

---

> > ### Author Response · Authors · 2025-11-28
> >
> > We sincerely thank the reviewer for the positive feedback. Following your suggestion, we have explicitly stated in the captions of Figure 17 and Figure 18, as well as in the text of Appendix A.4, that these presented results are randomly sampled. We appreciate your careful reading and constructive guidance. Thank you again for your support and encouraging comments.

---

### Official Review · Reviewer_vztR · 2025-11-03

**Soundness:** 3
**Presentation:** 3
**Contribution:** 3
**Rating:** 6
**Confidence:** 4

**Summary:**

This paper proposes a novel method, PartSAM, for 3D part segmentation. The high-level idea of PartSAM follows that of 2D SAM or PointSAM — it trains a feedforward model that takes a point prompt as input and outputs point masks. Unlike recent 3D part segmentation methods that leverage supervision from 2D SAM, this model is among the first to utilize 3D-native part information embedded in the raw mesh.

**Strengths:**

1. The paper addresses an important task, and the proposed method is well-motivated.

2. The idea of leveraging scene graphs and “model-in-the-loop annotation” is clever.

3. The paper is generally well written and easy to follow.

4. The proposed method outperforms previous methods.

5. The authors provide insightful ablation studies in the supplementary materials.

**Weaknesses:**

1. **Method novelty and comparison with existing work.**
   From a methodological standpoint, the high-level idea of the proposed model and training scheme closely follows that of 2D SAM, and therefore lacks substantial novelty or conceptual breakthrough. Moreover, similar attempts to extend 2D SAM to 3D already exist (e.g., PointSAM). Can the authors train PointSAM on the newly collected 3D dataset? Would it yield better or worse results? What are the specific advantages of the proposed network architecture or training scheme compared to PointSAM or 2D SAM?

2. **Contribution of large-scale 3D native data.**
   I understand that one of the main contributions of this paper lies in training on a large-scale 3D-native dataset. However, I would like to see a deeper quantitative and qualitative analysis of the data quality and its impact on model performance. For example, what are the concrete benefits of 3D-native data compared to 2D-lifted data? Is it more accurate, or does it offer other advantages such as better generalization or consistency? It would also be helpful if the authors could provide more discussion on whether the current data pipeline can reliably produce high-quality part annotations, and what remaining challenges exist in the data curation process.

3. **Experiments on data scale and quality.**
   I recommend adding further experiments to investigate the influence of data quantity and quality, such as scaling curves showing performance variations with different training data sizes.

4. **Impact of connectivity information.**
   I understand that connectivity information may play an important role in part segmentation. The current experiments only report results without considering connectivity information and appear to adapt the baselines accordingly. For a more comprehensive comparison, it would be beneficial to include both setups—*without* connectivity and *with* connectivity. In many scenarios, it is desirable for the model to utilize connectivity cues to produce more accurate and coherent part segmentations. Such comparisons would provide valuable insights into the model’s capability and potential advantages under different conditions.

**Questions:**

1. **Data release.**
   Will you release the data processing scripts and the processed datasets with part annotations to facilitate future research and ensure reproducibility?

2. **Use of connectivity information.**
   I noticed that the paper mentions the proposed method can leverage connectivity information “by applying graph cuts to the segmentation results for one iteration.” However, this description is still unclear. Could you please provide more details on how graph cuts are integrated into the pipeline?

3. **Hierarchical part decomposition.**
   Could you elaborate on how hierarchical part decomposition is achieved? Specifically, how are finer-grained parts derived from coarser-level segmentations, and what criteria or supervision signals are used to determine the part hierarchy?

4. **Runtime and efficiency.**
   Regarding the reported timing results, does the measurement correspond to the inference time for a single-point prompt, or to the runtime of the full automatic segmentation process that samples multiple point prompts? How many point prompts are typically used for automatic segmentation, and how does this number influence the overall runtime and efficiency?

---

> ### Author Response · Authors · 2025-11-23
>
> We thank the reviewer for their time in reviewing our paper and for their insightful comments. We are grateful that the reviewer found *the problem important, the proposed approach well-motivated, and the model-in-the-loop annotation strategy clever*. We also appreciate the positive remarks regarding *the clarity of the writing, the strong performance, and the insightful ablation studies*. Below, we tried our best to address the remaining concerns raised by the reviewer:
>
> > Method novelty and comparison with existing work
>
> **Training Point-SAM using our data:** To comprehensively compare with Point-SAM, we train the original Point-SAM model using our curated dataset. The significant performance drops on both the interactive and automatic segmentation tasks indicate that Point-SAM fails to scale on our native 3D data.
>
> | Method | IoU@1 | IoU@3 | IoU@5 | IoU@7 | Automatic |
> |----------|----------|----------|----------|----------|----------|
> | Point-SAM w/ Our Data  | 38.9  | 62.9  | 71.5  | 76.8  | 48.6  |
> | Ours  | **56.1**  | **79.0**  | **84.7**  | **86.9**  | **68.5**  |
>
> **Difference between Point-SAM and our method:** Compared with Point-SAM, our method offers advantages in both network architecture and training scheme for the part segmentation task. Regarding architecture, Point-SAM uses a transformer encoder operating on unstructured point clouds, whereas our method performs transformer modeling on a more advanced triplane feature field. The dual-branch encoder also brings in 2D SAM priors, which improve generalization and stabilize training on large-scale 3D data. We also benefit from the training scheme: the triplet contrastive loss strengthens part-aware feature learning, and its formulation is naturally robust to the scale and granularity ambiguity of 3D parts. Together, the continuous feature representation and the contrastive learning scheme give our framework substantially stronger scalability.
>
>
> > Scaling Analysis
>
> We conduct a scaling analysis in Appendix A.2.3 and report the scaling curve of PartSAM in Figure 11. The results show steady gains in both interactive and automatic segmentation as the training set grows, revealing a clear scaling law in 3D part segmentation. This demonstrates that PartSAM scales effectively with data size and our high-quality 3D-native annotations continue to provide consistent benefits with an increasing scale.
>
> >Qualitative Analysis of Training Data:
>
> In Figure 10, we compare our native 3D part labels with the 2D-lifting results generated via community detection in SAMesh. The lifted 2D masks are frequently incomplete and overly fragmented due to heavy occlusions, making them unsuitable as reliable supervision in our framework. In contrast, our native 3D annotations maintain strong spatial coherence and accurately capture interior structures. The advantage of precise interior part segmentation benefited by native 3D data is also illustrated in Figure 9.
>
> > Discussion on Reliable Part Annotation:
>
> Our dataset is constructed through a fully automated two-stage curation pipeline. We first filter out low-quality assets (e.g., scanned objects, abnormal part sizes, and shapes with too few or too many parts). We then refine the remaining data using high-poly artist meshes, connectivity-aware clustering, and a model-in-the-loop filtering process at both part and shape levels. Only shape–part pairs that pass all criteria are retained. This ensures high-quality 3D-native supervision at scale. Full details of the curation steps are provided in the revised paper.
> Our pipeline is fully automated to ensure scalability, so a small number of imperfect cases may remain, though their proportion is negligible. Importantly, the scaling curves in Figure 11 show consistent performance gains as data size increases, directly indicating that the overall data quality is sufficient and continues to benefit the model.
>
> >Remaining Challenges on Data Curation:
>
> The long-tail category problem in current data curation arises due to the class-agnostic nature of part annotation, where parts are annotated without predefined categories. Coupled with the high uncertainty inherent in open-world part segmentation—where part definitions are often ambiguous—this may result in an imbalanced distribution of parts in the dataset, making it difficult for the model to distinguish the rare, minority parts (failure cases in Figure 14).
>
> > Data Release
>
> We will release the data processing scripts and the processed datasets with part annotations to facilitate future research and ensure reproducibility

---

> ### Author Response · Authors · 2025-11-23
>
> > Use of connectivity information
>
> **Detail:** We refine the per-face labels using a standard graph-cut post-processing step. The mesh is modeled as a face-adjacency graph, and the algorithm minimizes an energy function composed of a data term (preserving the initial predictions) and a smoothness term (penalizing abrupt label changes between neighboring faces). More details are provided in Appendix A.2.5.
> **Comparison under this setting:** Comparison under both settings—with and without connectivity—have already been reported in Appendix A.2.5. Under the connectivity-enabled setting, PartSAM still achieves the best performance among all compared methods.
>
> > Hierarchical part decomposition
>
> **Hierachical mechanism:** In our method, hierarchical part decomposition does not come from refining coarse parts into fine ones. The multi-scale behavior directly results from the SAM-style parallel decoder, which naturally outputs masks at different granularities. For fair comparison, we only adjust the NMS threshold to select non-overlapping masks so that each mesh face receives a single part label. Changing this threshold yields different segmentation scales: a higher threshold keeps more masks in the same region, producing finer results. Notably, since PartSAM directly outputs masks at multiple scales, hierarchical segmentation can also be obtained through other simple controls, such as restricting the mask size before NMS. This means that our method can achieve scale-controllable segmentation similar to SAMPart3D.
> **Supervision signals:** In our approach, we do not explicitly define a part hierarchy during training. Instead, the hierarchy emerges from the following components: (1) a large-scale dataset of part labels across various scales, (2) parallel decoding strategy similar to SAM, and (3) a scale-invariant triplet contrastive loss used to train the field-based encoder. These elements collectively enable the model to learn parts at different levels of granularity.
>
> > Runtime and efficiency
>
> In the original manuscript, we reported the time for fully automatic segmentation, where 1,024 prompt points were sampled for each shape. We now provide a detailed breakdown of the runtime for each network component, as well as the performance under different numbers of prompt points. We observe that PartSAM maintains stable performance across varying prompt point counts, demonstrating its robustness to the choice of prompt density.
>
> | Time | Performance |
> |----------|----------|
> | ~12s (Encoder: 1.2s, Decoder: 0.01s × 1024 prompt points)  | 69.5  |
> | ~9s (768 prompt points)  | 68.8  |
> | ~7s (512 prompt points)  | 67.9  |
> | ~4s (256 prompt points)  | 65.2  |

---

### Author Response · Authors · 2025-11-23

We sincerely thank all reviewers for their constructive and detailed feedback. We have carefully addressed every concern in our point-by-point responses, and have revised the manuscript accordingly. All changes made during the rebuttal stage are highlighted in blue in the updated draft. We greatly appreciate the reviewers’ efforts and insightful comments, which have helped us substantially improve the clarity and quality of the paper.

---

### Author Response · Authors · 2025-12-01
**Summary of Rebuttal**

We thank the area chair for overseeing the review process, and we thank the reviewers for their thoughtful and constructive feedback. Across the reviews, there is broad agreement that this work makes a strong and well-executed contribution to scalable 3D part segmentation. Reviewers highlighted that our method delivers robust and fine-grained part segmentation with strong generalization—particularly on interior structures that prior approaches fail to capture—and that the pipeline is well-motivated, impactful, and supported by clear presentation and rigorous experiments.

During the rebuttal, we addressed all major concerns, including:
- **Additional Comparisons:** We conducted a comparison with Point-SAM trained on our curated native 3D data and showed that our method achieves substantially stronger scalability. These improvements arise from our architectural design and training scheme, which are specifically developed for scalable 3D part segmentation and are fundamentally different from prior approaches (**Appendix A.2.1**). We also extended the evaluation to the 3DCOMPAT++ dataset (**Appendix A.3**). In addition, we provided more visual comparisons on interior parts (**Figure 9**) and demonstrated that PartSAM exhibits stronger cross-shape consistency than PartField (**Appendix A.2.7**).
- **Data and Scalability Concerns:** We clarified the details of the native 3D dataset construction and the filtering process used to remove unreliable annotations (**Appendix A.1.3**). We also demonstrated that our method scales effectively under this setting (**Appendix A.2.3**). All data and code will be released in a fully license-compliant manner to ensure transparency and reproducibility.
- **Clarification of Technical Points:** We included complete details of the overall segmentation pipeline, the mechanism for incorporating previous prompts, and the IoU prediction head (**Appendix A.1.1 and A.1.2**). We then detailed how our method benefits from mesh connectivity and achieves higher performance than baselines under this setting (**Appendix A.2.5**). We further clarified how hierarchical part decomposition is realized in our framework and how it differs from prior approaches. In addition, we provided a more detailed runtime comparison and demonstrated that PartSAM is robust to variations in prompt density (**Appendix A.2.6**).
- **Additional Visualizations and Analyses:** We present failure cases of PartSAM in **Figure 14**, along with additional randomly sampled results on diverse meshes in **Figures 17 and 18**. We also provided a detailed analysis of the current limitations (**Appendix A.2.8**).

Due to the OpenReview incident, follow-up exchanges with some reviewers were  limited. Nevertheless, all reviewers who were able to engage during the rebuttal confirmed that their concerns were fully resolved. One reviewer updated their score from **4→6**, and another expressed an intention to raise their score from **6→8**. Based on the available updates, the post-rebuttal ratings stand at a minimum of **(8, 6, 6, 6)**.

---

### Meta-Review · Area_Chair_Dibo · 2026-01-06

**Summary:**

This paper introduces PartSAM, a novel 3D part segmentation model trained on large-scale 3D data. Compared to prior approaches that lift 2D segmentation to 3D, a model trained directly on 3D segmented shapes can better handle complex geometry and segment interior regions of 3D shapes. The overall framework is largely inspired by the SAM architecture for images. One key contribution is the curation of large-scale training data, involving model-in-the-loop annotation. The combination of a scalable architecture and diverse 3D training data enables open-vocabulary, promptable segmentation, which is a unique advantage of the proposed method over previous 3D segmentation models.

**Reviewer Concerns:**

Reviewers raised concerns about:
        1.        Lack of technical novelty (similarity to existing work such as PointSAM)
        2.        The advantages of 2D-lifting methods (e.g., SAMesh)
        3.        The impact of training data scale
        4.        The impact of connectivity information
        5.        Lack of failure cases
        6.        Lack of evaluation on a larger dataset such as 3DCOMPAT++
        7.        Insufficient technical details in the explanation

Overall, these concerns seemed to be addressed well in the rebuttal, supported by additional experiments and results.

**Reviewer Scores:**

The submission initially received scores of 6, 6, 4, and 6. After the rebuttal, two reviewers raised their scores, resulting in final scores of 6, 8, 6, and 6. All reviewers gave positive scores, and the AC also agrees with accepting the submission.

---

### Decision · Program_Chairs · 2026-01-26

Accept (Poster)